# Improving Environment Novelty Quantification for Effective Unsupervised Environment Design

**Jayden Teoh**[*], **Wenjun Li**[*†], **Pradeep Varakantham**
Singapore Management University
{jxteoh.2023, wjli.2020, pradeepv}@smu.edu.sg

## Abstract

Unsupervised Environment Design (UED) formalizes the problem of autocurricula through interactive training between a teacher agent and a student agent. The teacher generates new training environments with high learning potential, curating an adaptive curriculum that strengthens the student's ability to handle unseen scenarios. Existing UED methods mainly rely on *regret*, a metric that measures the difference between the agent's optimal and actual performance, to guide curriculum design. Regret-driven methods generate curricula that progressively increase environment complexity for the student but overlook environment *novelty*–a critical element for enhancing an agent's generalizability. Measuring environment novelty is especially challenging due to the underspecified nature of environment parameters in UED, and existing approaches face significant limitations. To address this, this paper introduces the *Coverage-based Evaluation of Novelty In Environment* (CENIE) framework. CENIE proposes a scalable, domain-agnostic, and curriculum-aware approach to quantifying environment novelty by leveraging the student's state-action space coverage from previous curriculum experiences. We then propose an implementation of CENIE that models this coverage and measures environment novelty using Gaussian Mixture Models. By integrating both regret and novelty as complementary objectives for curriculum design, CENIE facilitates effective exploration across the state-action space while progressively increasing curriculum complexity. Empirical evaluations demonstrate that augmenting existing regret-based UED algorithms with CENIE achieves state-of-the-art performance across multiple benchmarks, underscoring the effectiveness of novelty-driven autocurricula for robust generalization.

## 1 Introduction

Although recent advancements in Deep Reinforcement Learning (DRL) have led to many successes, e.g., super-human performance in games [26, 9] and reliable control in robotics [2, 3], training generally-capable agents remains a significant challenge. DRL agents often fail to generalize well to environments only slightly different from those encountered during training [21, 63]. To address this problem, there has been a surge of interest in *Unsupervised Environment Design* (UED; [56, 23, 57, 28, 27, 37, 33, 7]), which formulates the autocurricula [32] generation problem as a two-player zero-sum game between a *teacher* and a *student* agent. In UED, the teacher constantly adapts training environments (e.g., mazes with varying obstacles and car-racing games with different track designs) in the curriculum to improve the student's ability to generalize across all possible levels.

To design effective autocurricula, researchers have proposed various metrics to capture learning potential, enabling teacher agents to create training levels that adapt to the student's capabilities.

---

[*]Equal contribution.
[†]Corresponding author.

38th Conference on Neural Information Processing Systems (NeurIPS 2024).

The most popular metric, *regret*, measures the student's maximum improvement possible in a level. While regret-based UED algorithms [23, 27, 28] are effective in producing levels at the frontier of the student's capability, they do not guarantee diversity in the student's experiences, limiting the training of generally-capable agents especially in large environment design spaces. Another line of work in UED recognizes this limitation, leading to methods exploring the prioritization of novel levels during curriculum generation [56, 57, 33]. This strategic shift empowers the teacher to introduce novel levels into the curriculum such that the student agent can actively explore the environment space and enhance its generalization capabilities.

To more effectively evaluate environment novelty, we introduce the *Coverage-based Evaluation of Novelty In Environment* (CENIE) framework. CENIE operates on the intuition that a novel environment should induce unfamiliar experiences, pushing the student agent into unexplored regions of the state space and introducing variability in its actions. Therefore, signals about an environment's novelty can be derived by modeling and comparing its state-action space coverage with those of environments already encountered in the curriculum. We refer to this method of estimating novelty based on the agent's past experiences as *curriculum-aware*. By evaluating novelty in relation to the experiences induced by other environments within the curriculum, CENIE prevents redundant environments—those that elicit similar experiences as existing ones—from being classified as novel. Curriculum-aware approaches ensure that levels in the student's curriculum collectively drive the agent toward novel experiences in a sample-efficient manner.

Our contributions are threefold. First, we introduce CENIE, a scalable, domain-agnostic, and curriculum-aware framework for quantifying environment novelty via the agent's state-action space coverage. CENIE addresses shortcomings in existing methods for environment novelty quantification, as discussed further in Sections 3 and 4. Second, we present implementations for CENIE using *Gaussian Mixture Models* (GMM) and integrated its novelty objective with PLR$^\perp$[28] and ACCEL[37], the leading UED algorithms in zero-shot transfer performance. Finally, we conduct a comprehensive evaluation of the CENIE-augmented algorithms across three distinct benchmark domains. By incorporating CENIE into these leading UED algorithms, we empirically validate that CENIE's novelty-based objective not only exposes the student agent to a broader range of scenarios in the state-action space, but also contributes to achieving state-of-the-art zero-shot generalization performance. This paper underscores the importance of novelty and the effectiveness of the CENIE framework in enhancing UED.

## 2 Background

We briefly review the background of Unsupervised Environment Design (UED) in this section. UED problems are modeled as an Underspecified Partially Observable Markov Decision Process (UPOMDP) defined by the tuple:

$$\langle S, A, O, \mathcal{I}, \mathcal{T}, \mathcal{R}, \gamma, \Theta \rangle$$

where $S$, $A$ and $O$ are the sets of states, actions, and observations, respectively. $\Theta$ represents a set of free parameters where each $\theta \in \Theta$ defines a specific instantiation of an environment (also known as a *level*). We use the terms "environments" and "levels" interchangeably throughout this paper. The level-conditional observation and transition functions are defined as $\mathcal{I} : S \times \Theta \to O$ and $\mathcal{T} : S \times A \times \Theta \to \Delta(S)$, respectively. The student agent, with policy $\pi$, receives a reward based on the reward function $\mathcal{R} : S \times A \to \mathbb{R}$ with a discount factor $\gamma \in [0, 1]$. The student seeks to maximize its expected value for each $\theta$ denoted by $V^\theta(\pi) = \mathbb{E}_\pi[\sum_{t=0}^{T} R(s_t, a_t)\gamma^t]$. The teacher's goal is to select levels forming the curriculum by maximizing a utility function $U(\pi, \theta)$, which depends on $\pi$.

Different UED approaches vary primarily in the teacher's utility function. *Domain Randomization* (DR; [53]) uniformly randomizes environment parameters, with a constant utility $U^\mathcal{U}(\pi, \theta) = C$, making it agnostic to the student's policy. *Minimax training* [39] adversarially generates challenging levels, with utility $U^\mathcal{M}(\pi, \theta) = -V^\theta(\pi)$, to minimize the student's return. However, this approach incentivizes the teacher to make the levels completely unsolvable, limiting room for learning. Recent UED methods address this by using a teacher that maximizes *regret*, defined as the difference between the return of the optimal policy and the current policy. Regret-based utility is defined as $U^\mathcal{R}(\pi, \theta) = \text{REGRET}^\theta(\pi, \pi^*) = V^\theta(\pi^*) - V^\theta(\pi)$ where $\pi^*$ is the optimal policy on $\theta$. Regret-based objectives have been shown to promote the simplest levels that the student cannot solve optimally, and benefit from the theoretical guarantee of a minimax regret robust policy upon convergence in the

two-player zero-sum game. However, since $\pi^*$ is generally unknown, regret must be approximated. Dennis et al. [23], the pioneer UED work, introduced a principled level generation based on the regret objective and proposed the *PAIRED* algorithm, where regret is estimated by the difference between the returns attained by an antagonist agent and the protagonist (student) agent. Later on, Jiang et al. [27] introduced *PLR$^\perp$* which combines DR with regret using *Positive Value Loss* (PVL), an approximation of regret based on Generalized Advantage Estimation (GAE; [48]):

$$\text{PVL}^\theta(\pi) = \frac{1}{T} \sum_{t=0}^{T} \max \left( \sum_{k=t}^{T} (\gamma\lambda)^{k-t} \delta_k^\theta, 0 \right), \tag{1}$$

where $\lambda$ and $T$ are the GAE discount factor and MDP horizon, respectively. $\delta_k^\theta$ is the TD-error at time step $k$ for $\theta$. The state-of-the-art UED algorithm, *ACCEL* [37], improves PLR$^\perp$ [27] by replacing its random level generation with an editor that mutates previously curated levels to gradually introduce complexity into the curriculum.

## 3  Related Work

It is important to note that regret-based UED approaches provide a minimax regret guarantee at Nash Equilibrium; however, they provide no explicit guarantee of convergence to such equilibrium. Beukman et al. [10] demonstrated that the minimax regret objective does not necessarily align with learnability: an agent may encounter UPOMDPs with high regret on certain levels where it already performs optimally (given the partial observability constraints), while there exist other levels with lower regret where it could still improve. Consequently, selecting levels solely based on regret can lead to *regret stagnation*, where learning halts prematurely. This suggests that focusing exclusively on minimax regret may inhibit the exploration of levels where overall regret is non-maximal, but opportunities for acquiring transferable skills for generalization are significant. Thus, there is a compelling need for a complementary objective, such as novelty, to explicitly guide level selection towards enhancing zero-shot generalization performance and mitigating regret stagnation.

The *Paired Open-Ended Trailblazer* (POET; [56]) algorithm computes novelty based on environment encodings—a vector of parameters that define level configurations. POET maintains a record of the encodings from previously generated levels and computes the novelty of a new level by measuring the average distance between the k-nearest neighbors of its encoding. However, this method for computing novelty is domain-specific and relies on human expertise in designing environment encodings, posing challenges for scalability to complex domains. Moreover, due to UED's underspecified nature, where free parameters may yield a one-to-many mapping between parameters and environments instances, each inducing distinct agent behaviors, quantifying novelty based on parameters alone is futile.

*Enhanced POET* (EPOET; [57]) improves upon its predecessor by introducing a domain-agnostic approach to quantify a level's novelty. EPOET is grounded in the insight that novel levels offer new insights into how the behaviors of agents within them differ. EPOET evaluates both active and archived agents' performance in each environment, converting their performance rankings into rank-normalized vectors. The level's novelty is then computed by measuring the Euclidean distance between these vectors. Despite addressing POET's domain-specific limitations, EPOET encounters its own challenges. The computation of rank-normalized vectors only works for population-based approaches as it requires evaluating multiple trained student agents and incurs substantial computational costs. Furthermore, EPOET remains curriculum-agnostic, as its novelty metric relies on the ordering of raw returns within the agent population, failing to directly assess whether the environment elicits rarely observed states and actions in the existing curriculum.

*Diversity Induced Prioritized Level Replay* (DIPLR; [33]), calculates novelty using the Wasserstein distance between occupancy distributions of agent trajectories from different levels. DIPLR maintains a level buffer and determines a level's novelty as the minimum distance between the agent's trajectory on the candidate level and those in the buffer. While DIPLR incorporates the agent's experiences into its novelty calculation, it faces significant scalability and robustness issues. First, relying on the Wasserstein distance is notoriously costly. Additionally, DIPLR requires pairwise distance computations between all levels in the buffer, causing computational costs to grow exponentially with more levels. Finally, although DIPLR promotes diversity within the active buffer, it fails to evaluate whether state-action pairs in the current trajectory have already been adequately explored through

past curriculum experiences, making it arguably still curriculum-agnostic. Further discussions on relevant literature can be found in Appendix B.

## 4  Approach: CENIE

The limitations of prior approaches to quantifying environment novelty underscore the need for a more robust framework, motivating the development of CENIE. CENIE quantifies environment novelty through state-action space coverage derived from the agent's accumulated experiences across previous environments in its curriculum. In single-environment online reinforcement learning, coverage within the training distribution is often linked to sample efficiency [59], providing inspiration for the CENIE framework. Given UED's objective to enhance a student's generalizability across a vast and often unseen (during training) environment space, quantifying novelty in terms of state-action space coverage has several benefits. By framing novelty in this way, CENIE enables a sample-efficient exploration of the environment search space by prioritizing levels that drive the agent towards unfamiliar state-action combinations. This provides a principled basis for directing the environment design towards enhancing the generalizability of the student agent. Additionally, a distinctive benefit of this approach is that it is not confined to any particular UED or DRL algorithms since it solely involves modeling the agent's state-action space coverage. This flexibility allows us to implement CENIE atop any UED algorithm.

CENIE's approach to novelty quantification through state-action coverage introduces three key attributes, effectively addressing the limitations of previous methods: (1) **domain-agnostic**, (2) **curriculum-aware**, and (3) **scalable**. CENIE is domain-agnostic, as it quantifies novelty solely based on the state-action pairs of the student, thus eliminating any dependency on the encoding of the environment. CENIE achieves curriculum-awareness by quantifying novelty using a model of the student's past aggregated experiences, i.e., state-action space coverage, ensuring that the selection of environments throughout the curriculum is sample-efficient with regards to expanding the student's state-action coverage. Lastly, CENIE demonstrates scalability by avoiding the computational burden associated with exhaustive pairwise comparisons or costly distance metrics.

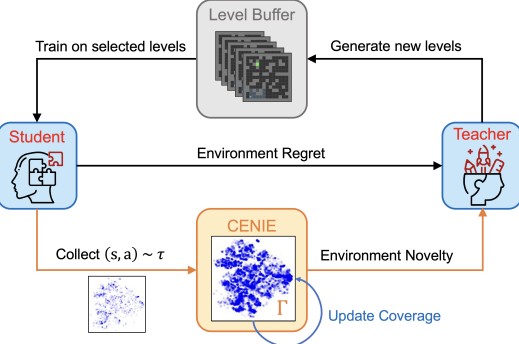

Figure 1: An overview of the CENIE framework. The teacher will utilise environment regret and novelty for curating student's curriculum. $\Gamma$ contains past experiences and $\tau$ is the recent trajectory.

### 4.1  Measuring the Novelty of a Level

To evaluate the novelty of new environments using the agent's state-action pairs, the teacher needs to first model the student's past state-action space coverage distribution. We propose to use GMMs as they are particularly effective due to their robustness in representing high-dimensional continuous distributions [14, 5]. A GMM is a probabilistic clustering model that represents the underlying distribution of data points using a weighted combination of multivariate Gaussian components. Once the state-action distribution is modeled using a GMM, we can leverage it for density estimation. Specifically, the GMM allows us to evaluate the likelihood of state-action pairs induced by new environments, where lower likelihoods indicate experiences that are less represented in the student's current state-action space. This likelihood provides a quantitative measure of dissimilarity in state-action space coverage, enabling a direct comparison of novelty between levels. It is important to note that CENIE defines a general framework for quantifying novelty through state-action space coverage;

GMMs represent just one possible method for modeling this coverage. Future research may explore alternatives to model state-action space coverage within the CENIE framework (see Section C in the appendix for more discussions).

Before detailing our approach, we first define the notations used in this section. Let $l_\theta$ be a particular level conditioned by an environment parameter $\theta$. We refer to $l_\theta$ as the candidate level, for which we aim to determine its novelty. The agent's trajectory on $l_\theta$ is denoted as $\tau_\theta$, and can be decomposed into a set of sample points, represented as $X_\theta = \{x = (s, a) \sim \tau_\theta\}$. The set of past training levels is represented by $L$ and $\Gamma = \{x = (s, a) \sim \tau_L\}$ is a buffer containing the state-action pairs collected from levels across $L$. We treat $\Gamma$ as the ground truth of the agent's state-action space coverage, against which we evaluate the novelty of state-action pairs from the candidate level $X_\theta$.

To fit a GMM on $\Gamma$, we must find a set of Gaussian mixture parameters, denoted as $\lambda_\Gamma = \{(\alpha_1, \mu_1, \Sigma_1), ..., (\alpha_K, \mu_K, \Sigma_K)\}$, that best represents the underlying distribution. Here, $K$ denotes the predefined number of Gaussians in the mixture, where each Gaussian component is character-ized by its weight ($\alpha_k$), mean vector ($\mu_k$), and covariance matrix ($\Sigma_k$), with $k \in \{1, ..., K\}$. We employ the *kmeans++* algorithm [12, 4] for a fast and efficient initialization of $\lambda_\Gamma$. The likelihood of observing $\Gamma$ given the initial GMM parameters $\lambda_\Gamma$ is expressed as:

$$P(\Gamma \mid \lambda_\Gamma) = \prod_{j=1}^{J} \sum_{k=1}^{K} \alpha_k \mathcal{N}(x_j \mid \mu_k, \Sigma_k) \tag{2}$$

where $x_j$ is a state-action pair sample from $\Gamma$. $\mathcal{N}(x_j \mid \mu_k, \Sigma_k)$ represents the multi-dimensional Gaussian density function for the $k$-th component with mean vector $\mu_k$ and covariance matrix $\Sigma_k$. To optimise $\lambda_\Gamma$, we use the Expectation Maximization (EM) algorithm [22, 43] because Eq. 2 is a non-linear function of $\lambda_\Gamma$, making direct maximization infeasible. The EM algorithm iteratively refines the initial $\lambda_\Gamma$ to estimate a new $\lambda'_\Gamma$ such that $P(X \mid \lambda'_\Gamma) > P(X \mid \lambda_\Gamma)$. This process is repeated iteratively until some convergence, i.e., $\| \lambda'_\Gamma - \lambda_\Gamma \| < \epsilon$, where $\epsilon$ is a small threshold.

Once the GMM is fitted, we can use $\lambda_\Gamma$ to perform density estimation and calculate the novelty of the candidate level $l_\theta$. Specifically, we consider the set of state-action pairs from the agent's trajectory, $X_\theta$, and compute their posterior likelihood under the GMM. This likelihood indicates how similar the new state-action pairs are to the learned distribution of past state-action coverage. Consequently, the novelty score of $l_\theta$ is represented as follows:

$$\text{NOVELTY}_{l_\theta} = -\frac{1}{|X_\theta|} \log \mathcal{L}(X_\theta \mid \lambda_\Gamma) = -\frac{1}{|X_\theta|} \sum_{t=1}^{T} \log p(x_t \mid \lambda_\Gamma) \tag{3}$$

where $x_t$ is a sample state-action pair from $X_\theta$. As shown in Eq. 3, we take the negative mean log-likelihood across all samples in $X_\theta$ to attribute higher novelty scores to levels with state-action pairs that are less likely to originate from the aggregated past experiences, $\Gamma$. This novelty metric promotes candidate levels that induce more novel experiences for the agent during training. More details on fitting GMMs are explained in Appendix D.1.

**Design considerations for the GMM** First, we specifically designate the state-action coverage buffer, i.e., $\Gamma$, as a First-In-First-Out (FIFO) buffer with a fixed window length. By focusing on a fixed window rather than the entire history of state-action pairs, our novelty metric avoids bias toward experiences that are outdated and have not appeared in recent trajectories. This design choice helps reduce the effects of catastrophic forgetting prevalent in DRL. Next, it is known that by allowing the adaptation of the number of Gaussians in the mixture, i.e., $K$ in Eq. 2, any smooth density distribution can be approximated arbitrarily close [24]. Therefore, to optimize the GMM's representation of the agent's state-action coverage distribution, we fit multiple GMMs with varying numbers of Gaussians within a predefined range at each time step and select the best one based on the silhouette score [45], an approach inspired by Portelas et al. [40]. The silhouette score evaluates clustering quality by measuring both intra-cluster cohesion and inter-cluster separation. This approach enables CENIE to construct a pseudo-online GMM model that dynamically adjusts its complexity to accommodate the agent's changing state-action coverage distribution.

## 4.2 State-Action Space Coverage Directed Training Agent

**Algorithm 1** ACCEL-CENIE
***
**Input**: Level buffer size $N$, Component range $[K_{min}, K_{max}]$, FIFO window size $\mathcal{W}$, level generator $\mathcal{G}$
**Initialize**: Student policy $\pi_\eta$, level buffer $\mathcal{B}$, state-action buffer $\Gamma$, GMM parameters $\lambda_\Gamma$
 1: Generate $N$ initial levels by $\mathcal{G}$ to populate $\mathcal{B}$
 2: Collect $\pi_\eta$'s trajectories on each level in $\mathcal{B}$ and fill up $\Gamma$
 3: **while** not converged **do**
 4:     Sample replay decision, $\epsilon \sim U[0, 1]$
 5:     **if** $\epsilon \geq 0.5$ **then**
 6:         Generate a new level $l_\theta$ by $\mathcal{G}$
 7:         Collect trajectories $\tau$ on $l_\theta$, with stop-gradient $\eta_\perp$
 8:         Compute novelty score for $l_\theta$ using $\lambda_\Gamma$ (Eq.3 and Eq.4)
 9:         Compute regret score for $l'_\theta$ (Eq.1 and Eq.4)
10:         Update $\mathcal{B}$ with $l_\theta$ if $P_{replay}(l_\theta)$ is greater than that of any levels in $\mathcal{B}$ (Eq.5)
11:     **else**
12:         Sample a replay level $l_\theta \sim \mathcal{B}$ according to $P_{replay}$
13:         Collect trajectories $\tau$ on $l_\theta$
14:         Update $\pi_\eta$ with rewards $R(\tau)$
15:         Update $\Gamma$ with $\tau$ and resize to $\mathcal{W}$
16:         **for** $k$ in range $K_{min}$ to $K_{max}$ **do**
17:            Fit a $GMM_k$ with $k$ components on $\Gamma$ and compute its silhouette score
18:         **end for**
19:         Select GMM parameters with the highest silhouette score to replace $\lambda_\Gamma$
20:         Perform edits on $l_\theta$ to produce $l'_\theta$
21:         Collect trajectories $\tau$ on $l'_\theta$, with stop-gradient $\eta_\perp$
22:         Compute novelty score for $l'_\theta$ using $\lambda_\Gamma$ (Eq.3 and Eq.4)
23:         Compute regret score for $l'_\theta$ (Eq.1 and Eq.4)
24:         Update $\mathcal{B}$ with $l'_\theta$ if $P_{replay}(l'_\theta)$ is greater than that of any levels in $\mathcal{B}$ (Eq.5)
25:     **end if**
26: **end while**
***

With a scalable method to quantify the novelty of levels, we demonstrate its versatility and effectiveness by deploying it on top of the leading UED algorithms, PLR$^\perp$ and ACCEL. For convenience, in subsequent sections, we will refer to this CENIE-augmented methodology of PLR$^\perp$ and ACCEL using GMMs as PLR-CENIE and ACCEL-CENIE, respectively. Both PLR$^\perp$ and ACCEL utilize a replay mechanism to train their students on the highest-regret levels curated within the level buffer. To integrate CENIE within these algorithms, we use normalized outputs of a prioritization function to convert the level scores (novelty and regret) into level replay probabilities ($P_S$):

$$P_S = \frac{h(S_i)^\beta}{\sum_j h(S_j)^\beta} \tag{4}$$

where $h$ is a prioritization function (e.g. rank) with a tunable temperature $\beta$ that defines the prioritization of levels with regards to any arbitrary score $S$. Following the implementations in PLR$^\perp$ and ACCEL, we employ $h$ as the rank prioritization function, i.e., $h(S_i) = 1/\text{rank}(S_i)$, where $\text{rank}(S_i)$ is the rank of level score $S_i$ among all scores sorted in descending order. In ACCEL-CENIE and PLR-CENIE, we use both the novelty and regret scores to determine the level replay probability:

$$P_{replay} = \alpha \cdot P_N + (1 - \alpha) \cdot P_R \tag{5}$$

where $P_N$ and $P_R$ are the novelty-prioritized probability and regret-prioritized probability respectively, and $\alpha$ allows us to adjust the weightage of each probability. The complete procedures for ACCEL-CENIE are provided in Algorithm 1, and for PLR-CENIE in the appendix (see Algorithm 2). Key steps specific to CENIE using GMMs are highlighted in blue.

## 5 Experiments

In this section, we benchmark PLR-CENIE and ACCEL-CENIE against their predecessors and a set of baseline algorithms: Domain Randomization (DR), Minimax, PAIRED, and DIPLR. The

technical details of each algorithm are presented in Appendix E. We empirically demonstrated the effectiveness of CENIE on three distinct domains: Minigrid, BipedalWalker, and CarRacing. Minigrid is a partially observable navigation task under discrete control with sparse rewards, while BipedalWalker and CarRacing are partially observable continuous control tasks with dense rewards. Due to the complexity of mutating racing tracks, CarRacing is the only domain where ACCEL and ACCEL-CENIE are excluded. The experiment details are provided in Appendix D. Following standard UED practices, all agents were trained using Proximal Policy Optimization (PPO; [49]) across the domains, and we present their zero-shot performance on held-out tasks. We also conducted ablation studies to assess the isolated effectiveness of CENIE's novelty metric (see Appendix A).

For reliable comparison, we employ the standardized DRL evaluation metrics [1], presenting the aggregate inter-quartile mean (IQM) and optimality gap plots after normalizing the performance with a min-max range of solved-rate/returns. Specifically, IQM focuses on the middle 50% of combined runs, discarding the bottom and top 25%, thereby providing a robust measure of overall performance. Optimality gap captures the amount by which the algorithm fails to meet a desirable target (e.g., 95% solved rate), beyond which further improvements are considered unimportant. Higher IQM and lower optimality gap scores are better. The hyperparameters for the algorithms in each experiment are specified in the appendix.

## 5.1 Minigrid Domain

First, we validated the CENIE-augmented methods in Minigrid [23, 20], a popular benchmark due to its ease of interpretability and customizability. Given its sparse reward signals and partial observability, navigating Minigrid requires the agent to explore multiple possible paths before successfully solving the maze and receiving rewards for policy updates. Therefore, Minigrid is an ideal domain to validate the exploration capabilities of the CENIE-augmented algorithms.

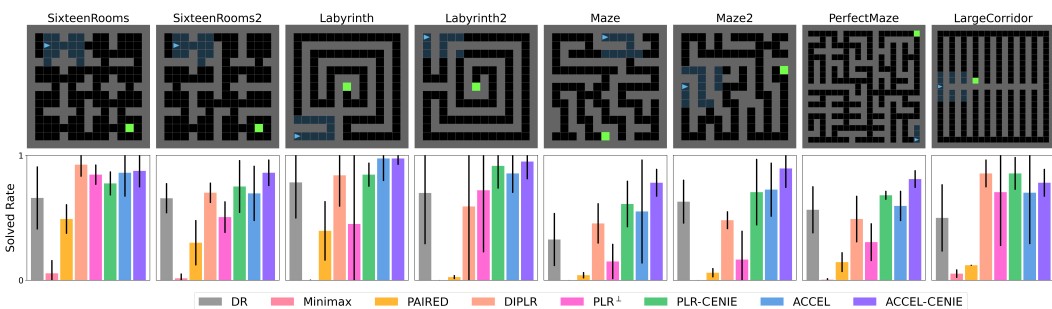

Figure 2: Zero-shot transfer performance in eight human-designed test environments. The plots are based on the median and interquartile range of solved rates across 5 independent runs.

Following prior UED works, we train all student agents for 30k PPO updates (approximately 250 million steps) and evaluate their generalization on eight held-out environments (see Figure 2). Figure 2 demonstrates that ACCEL-CENIE outperforms ACCEL in all testing environments. Moreover, PLR-CENIE shows significantly better performance in seven test environments compared to PLR$^\perp$. This underscores the ability of CENIE's novelty metric to complement the UED framework, particularly in improving generalization performance beyond the conventional learning potential metric, regret. Further empirical validation in Figure 3(a) confirms ACCEL-CENIE's superiority over ACCEL in both IQM and optimality gap. PLR-CENIE also outperforms its predecessor, PLR$^\perp$, by a significant margin. Notably, PLR-CENIE's performance is able to match ACCEL's, which is significant considering PLR-CENIE uses a random generator while ACCEL uses an editing mechanism to introduce gradual complexity to environments.

Beyond the normal-size testing mazes, we consider a more challenging evaluation setting. We evaluate the fully-trained student policy of PLR$^\perp$, PLR-CENIE, ACCEL, and ACCEL-CENIE on `PerfectMazeLarge` (shown in Figure 3(b)), an out-of-distribution environment which has $51 \times 51$ tiles and a episode length of 5000 timesteps – a much larger scale than training levels. We evaluate the agents for 100 episodes (per seed), using the same checkpoints in Figure 2. ACCEL-CENIE and ACCEL achieved comparable zero-shot transfer performance. Notably, PLR-CENIE achieved close

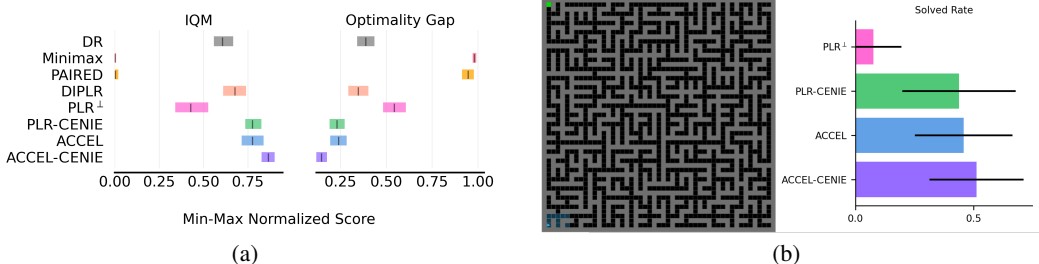

(a)               (b)

Figure 3: (a) Aggregate zero-shot transfer performance in Minigrid domain across 5 independent runs. (b) Zero-shot test performance of PLR$^\perp$, PLR-CENIE, ACCEL, and ACCEL-CENIE on PerfectMazeLarge across 5 independent runs.

to 50% solved rate, matching ACCEL's performance. This is a significant improvement from PLR$^\perp$, which had less than a 10% solved rate.

## 5.2 BipedalWalker Domain

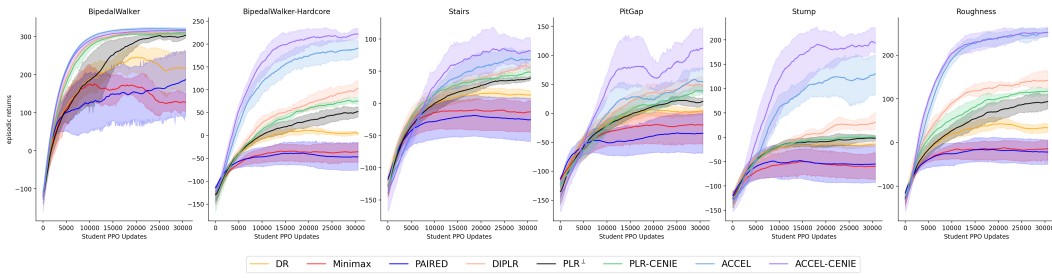

Figure 4: Student's generalization performance on 6 BipedalWalker testing environments during training. Each curve is measured across 5 independent runs (mean and standard error).

We also evaluated the CENIE-augmented algorithms in the BipedalWalker domain [56, 37], which is a partially observable continuous domain with dense rewards. We train all the algorithms for 30k PPO updates ($\sim$1B timesteps) and then evaluate their generalization performance on six distinct test environments: BipedalWalker (default), Hardcore, Stair, PitGap, Stump, and Roughness (visualized in Figure 6(a)). To monitor the student's generalization performance evolution, we assess the student policy every 100 PPO updates across six testing environments during the training period.

In Figure 4, ACCEL-CENIE outperforms ACCEL in five testing environments, with both achieving parity in the Roughness challenge, establishing ACCEL-CENIE as the leading UED algorithm in BipedalWalker. Similarly, PLR-CENIE consistently outperforms PLR$^\perp$ across all testing instances, except for the Stump challenge, where both algorithms exhibit similar performance. We present the aggregate results after min-max normalization (with range=[0, 300] on all test environments) in Figure 6(b). Both ACCEL-CENIE and PLR-CENIE exhibit better performance compared to their predecessors in the IQM and optimality gap metrics. Notably, ACCEL-CENIE outperforms all benchmarks by a substantial margin, achieving close to 55% of optimal performance.

Table 1: Coverage of state-action space across 30k PPO updates in the BipedalWalker domain.

|  | PLR$^\perp$ | PLR-CENIE | ACCEL | ACCEL-CENIE |
|---|---|---|---|---|
| State-action Space Coverage | 43.4% | 55.3% | 42.5% | 47.6% |

Next, we tracked the evolution of state-action space coverage throughout training to evaluate the impact of CENIE's novelty objective on the curriculum's exploration of the state-action space. During

training, state-action pairs encountered by the agent were collected for both PLR$^\perp$ and ACCEL, along with their CENIE-augmented versions. To visualize the distribution of these high-dimensional state-action pairs, we applied t-distributed Stochastic Neighbor Embedding (t-SNE; [54]) to project them into a 2-D space. The resulting evolution plot and detailed implementation steps are provided in Appendix A.2. Afterwards, we quantified state-action space coverage by discretizing the 2-D scatterplot into cells and calculating the percentage of total cells occupied by each algorithm. As shown in Table 1, CENIE drives ACCEL-CENIE and PLR-CENIE to achieve significantly broader state-action coverage by the end of 30k PPO updates compared to their predecessors. This evidence supports that the inclusion of CENIE's novelty objective for level replay prioritization contributes to broader state-action space coverage.

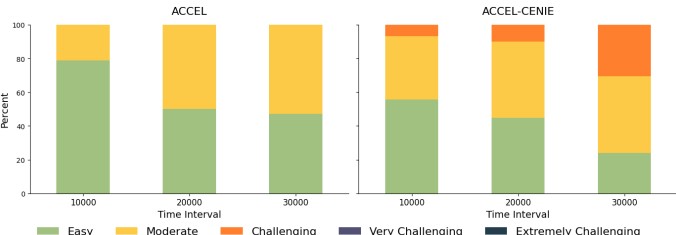

Figure 5: Difficulty composition of levels replayed by ACCEL and ACCEL-CENIE during training.

To understand ACCEL-CENIE's improvement over ACCEL, we analyzed the difficulty composition of replayed levels at various training intervals across five seeds, as shown in Figure 5. Level difficulty is assessed based on environment parameters such as stump height and pit gap width, using metrics adapted from Wang et al. [56] (details in Appendix A.2). It is evident that ACCEL predominantly favors "Easy" to "Moderate" difficulty levels, whereas ACCEL-CENIE progressively incorporates "Challenging" levels into its replay selection throughout training.

The disparity in level difficulty distribution between ACCEL and ACCEL-CENIE is a critical factor in understanding their observed performance differences. ACCEL's training curriculum tends to remain within a comfort zone, consistently selecting a limited subset of simpler levels where the agent experiences high regret. However, this can be problematic when considering the regret stagnation problem. Specifically, in the event where the easier levels exhibit *irreducible regret*, it can restrict the agent's exposure to more complex scenarios, thereby constraining its generalization potential. In contrast, ACCEL-CENIE's integration of a novelty objective actively selects challenging levels, pushing the agent beyond its comfort zone into unfamiliar, complex environments. This novelty-based regularization fosters the exploration of under-explored regions in the state-action space, even if regret levels are low, thereby enhancing the agent's generalization capabilities. Furthermore, with a mutation-based approach like ACCEL, this environment selection strategy may generate or mutate new levels with high learning potential, further enriching the training curriculum.

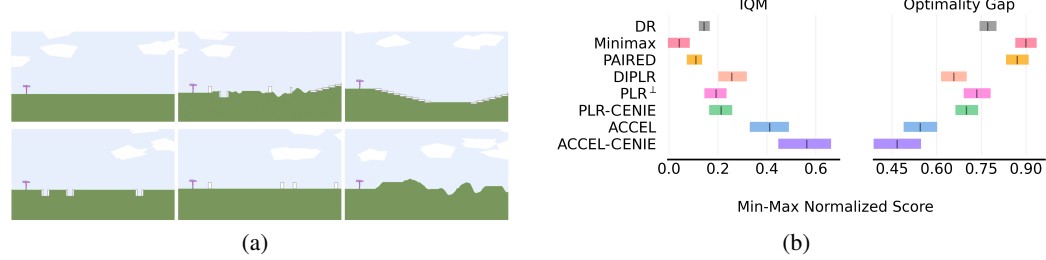

(a)                                                                              (b)

Figure 6: (a) BipedalWalker domain and (b) Aggregate zero-shot transfer performance in Bipedal-Walker.

## 5.3 CarRacing Domain

Finally, we evaluated the effectiveness of CENIE by implementing it on PLR$^\perp$ within the `CarRacing` domain [16, 27]. In this domain, the teacher manipulates the curvature of racing tracks using Bézier

curves defined by a sequence of 12 control points, while the student drives on the track under continuous control with dense rewards. We train the students in each algorithm for 2.75k PPO updates (∼5.5M steps), after which we test the zero-shot transfer performance of the different algorithms on 20 levels replicating real-world Formula One (F1) tracks introduced by Jiang et al. [27]. These tracks are guaranteed to be OOD as their configuration cannot be defined by Bézier curves with only 12 control points. The middle image in Figure 6b shows a track generated by domain randomization and the rightmost image shows a bird's-eye view of the F1-USA benchmark track.

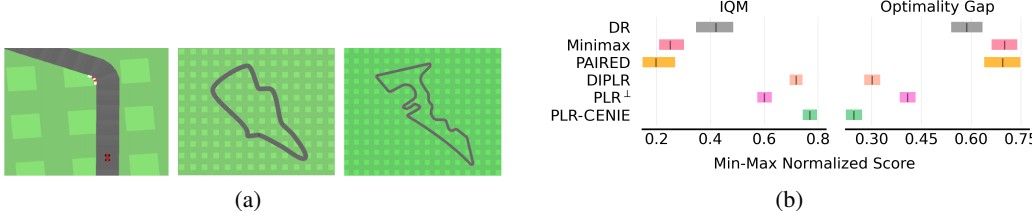

(a)                                                     (b)

Figure 7: (a) CarRacing domain and (b) Aggregate zero-shot transfer performance in CarRacing.

The aggregate performance after min-max normalization of all algorithms is summarized in Figure 7b. Note that the min-max range varies across F1 tracks due to different specifications on the maximum episode steps (see Table 5 in the appendix for more details). Once again, the CENIE-augmented algorithm, PLR-CENIE, achieves the best generalization performance in both IQM and optimality gap scores. Table 3 in the appendix shows the zero-shot transfer returns on all 20 F1 tracks. PLR-CENIE consistently outperforms or matches the best-performing baseline on all tracks.

Figure 8 presents the total regret in the level replay buffer for both PLR$^\perp$ and PLR-CENIE throughout the training process. Interestingly, PLR-CENIE maintains comparable, or even slightly higher, levels of regret across the training distribution, despite not directly optimizing for it. This outcome suggests that CENIE's novelty objective synergizes with the discovery of high-regret levels, providing counterintuitive evidence that optimizing solely for regret is not the only, nor necessarily the most effective, strategy for identifying levels with high learning potential (as approximated by regret). Intuitively, value predictions are inherently less reliable in regions of lower coverage density–areas characterized by higher entropy or high uncertainty regarding optimal actions–since these regions are less frequently sampled for agent's learning. These high-entropy regions are prime candidates for high-regret outcomes, especially when using a bootstrapped regret estimate, as in Eq. 1, due to the value estimation error in such states. By pursuing novel environments based on coverage, CENIE indirectly enhances the discovery of high-regret states, highlighting that novelty-driven autocurricula can effectively complement regret-based methods in uncovering diverse and challenging training scenarios.

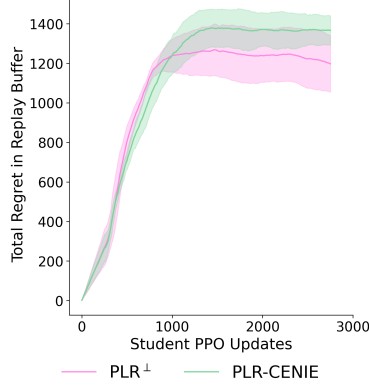

Figure 8: Total regret in level replay buffer for PLR$^\perp$ and PLR-CENIE over training in CarRacing.

## 6    Conclusion

In this paper, we introduced Coverage-based Evaluation of Novelty In Environment (CENIE), a scalable, domain-agnostic, and curriculum-aware framework for quantifying environment novelty in UED. We then proposed an implementation of CENIE that models this coverage and measures environment novelty using Gaussian Mixture Models. By incorporating CENIE with existing UED algorithms, we validated the framework's effectiveness in enhancing agent exploration capabilities and zero-shot transfer performance across three distinct benchmark domains. This promising approach marks a significant step towards unifying novelty-driven exploration and regret-driven exploitation for training generally capable RL agents. We encourage motivated readers to refer to the appendix for further studies and discussions on CENIE.

## Acknowledgments

This research/project is supported by the National Research Foundation Singapore and DSO National Laboratories under the AI Singapore Programme (AISG Award No: AISG2-RP-2020-017) and Lee Kuan Yew Fellowship awarded to Pradeep Varakantham.

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

# A  Extended Experiment Details and Ablation Studies

In this section, we present extended experiment details regarding the results presented in the main body of the paper. To isolate the individual effects of regret and novelty, we conduct an ablation study in which only novelty is used to prioritize levels in the level buffer. We denote the CENIE-augmented versions of the $PLR^\perp$ and ACCEL, which use only novelty for level prioritization (set $\alpha = 1$ in Equation 5), as PLR-CENIE$^\dagger$ and ACCEL-CENIE$^\dagger$, respectively.

Interestingly, we observed that in several instances, the ablation models, PLR-CENIE$^\dagger$ and ACCEL-CENIE$^\dagger$, demonstrated comparable or even superior zero-shot transfer performance compared to their regret metric counterparts, $PLR^\perp$ and ACCEL. This finding suggests that, in specific scenarios, prioritizing training levels based on novelty alone can effectively shape curricula. This is especially notable because our GMM-based novelty metric, unlike regret, does not rely on predefined domain-specific reward structures; rather, it is derived solely from the agent's trajectory data across different levels.

However, it is important to note that these ablation results do not imply that regret should be entirely replaced by novelty-based level selection. Novelty alone may encounter limitations in extremely large state-action spaces where a balance with regret is essential for effective exploration. By combining novelty and regret in CENIE to shape the training curriculum, we significantly enhance the agent's generalization capabilities beyond those of previous algorithms, as shown in our main experiments. This finding highlights the powerful synergy between CENIE's novelty metric and traditional regret-based approaches, resulting in a more robust and effective training paradigm.

## A.1  Minigrid Domain

After training all the student agents for 30k PPO updates ($\sim$250M steps), we evaluate their transfer capabilities on eight held-out testing environments (see the first row in Figure 9). We summarize all the results in Figure 9. In addition to the zero-shot transfer evaluation, we summarize the students' aggregate zero-shot transfer performance, i.e., IQM and Optimality Gap, in Figure 10.

In Figure 9, PLR-CENIE$^\dagger$ outperforms $PLR^\perp$ in most of the testing environments (6 out of 8), indicating that the novelty metric plays a more significant role compared to regret in the Minigrid domain for the $PLR^\perp$ algorithm. In contrast, ACCEL shows a marginal performance advantage over ACCEL-CENIE$^\dagger$ in the testing environments, with two wins, four losses, and two ties. Importantly, for both cases – PLR-CENIE and ACCEL-CENIE – the combination of both regret and novelty yields the strongest performance, outperforming their individual metric counterparts. This finding supports the assertion that the CENIE framework effectively complements the regret metric, helping UED algorithms achieve better performance.

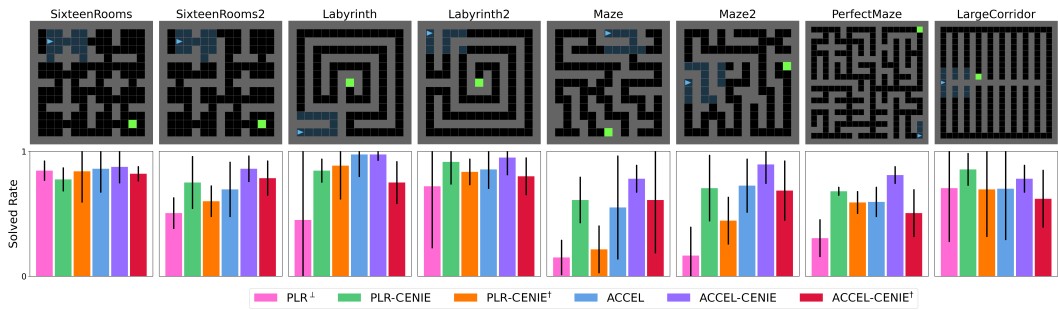

Figure 9: Zero-shot transfer performances in Minigrid. The plots are based on the median and interquartile range of solved rates across 5 independent runs. All student models are evaluated after 30k student PPO updates.

The aggregate IQM and Optimality Gap results shown in Figure 10 further validates the above conclusion. ACCEL-CENIE and PLR-CENIE outperform their counterparts – (ACCEL, ACCEL-CENIE$^\dagger$) and ($PLR^\perp$, PLR-CENIE$^\dagger$) – in terms of both IQM and Optimality Gap. In particular,

within the PLR$^\perp$ framework, the novelty-driven level selection strategy (PLR-CENIE$^\dagger$) significantly surpasses the regret-based approach (PLR$^\perp$) in performance.

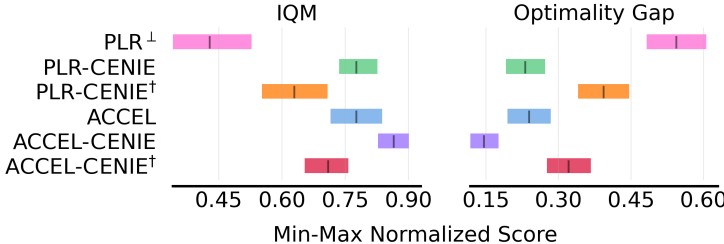

Figure 10: IQM and Optimality Gap ablations in Minigrid domain. Results are measured across 5 independent runs.

We also provide a qualitative analysis of the effect of the novelty metric on the level replay buffer of PLR-CENIE in Minigrid for the experiments detailed under Section 5.1 in the main body. Specifically, we highlight levels that feature the lowest regret (bottom 10) yet exhibit the highest novelty (top 10); these are showcased in the first row of Figure 11. Conversely, levels that score within the lowest 10 for both regret and novelty are displayed in the second row of the same figure. Visually, we observe that levels with high novelty and low regret present complex and diverse scenarios that challenge the student. In contrast, the levels displayed in the second row, characterized by low regret and low novelty, often resemble simple, empty mazes that offer limited learning opportunities. While it is not feasible to present every example level here, the contrast between the two groups is stark. Levels selected based on low regret but high novelty are significantly more varied and intricate than those chosen for their low novelty, despite both groups having low regret scores. This demonstrates that incorporating novelty alongside regret in the selection process enhances the ability to identify levels that present more interesting trajectories (experiences) for the student to learn from.

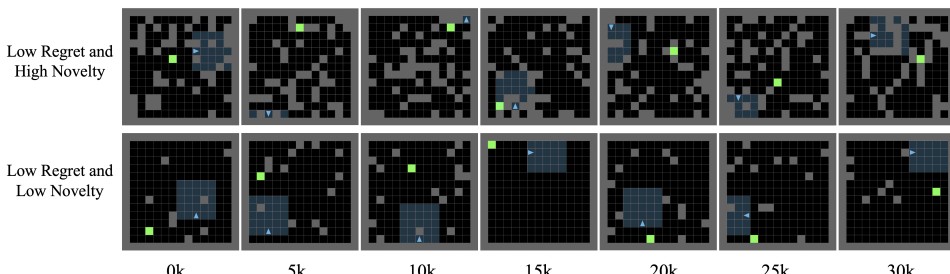

Figure 11: Levels in the level replay buffer of PLR-CENIE. X-axis: number of student PPO updates.

## A.2   BipedalWalker Domain

We closely tracked the evolution of state-action space coverage during the training to reveal how the incorporation of a novelty objective affected the curriculum generation. State-action pairs encountered by the agent during training are collected for PLR$^\perp$, ACCEL, PLR-CENIE, and ACCEL-CENIE. Given the high-dimensionality of the state-action pairs in the BipedalWalker domain, we employed t-distributed Stochastic Neighbor Embedding (t-SNE; [54]), a nonlinear dimensionality reduction technique, to project the state-action pairs onto a more manageable two-dimensional manifold. t-SNE captures much of the local structure of the high-dimensional data, while also revealing global structures, such as the presence of clusters at several scales [54, 58]. The resulting embedded state-action pairs are mapped onto a 2-D scatterplot, allowing us to visualize the exploration of the state-action space by each algorithm as the number of policy updates increases. The evolution is illustrated in Figure 12.

Furthermore, we quantified the occupancy of the 2-D scatterplot by each method. To achieve this, we discretized the scatterplot into cells and computed the percentage of total cells occupied by data

points generated by each method. Table 1 in the main paper presents the state-action space coverage percentages for each method. Notably, both PLR-CENIE and ACCEL-CENIE exhibit significantly broader coverage of the state-action space compared to their predecessors. This evidence supports the assertion that the outperformance of CENIE-augmented algorithms is associated with the broader coverage of the state-action space. Note that although the PLR-based algorithms exhibit higher state-action space coverage, they show poorer transfer performance compared to ACCEL-based algorithms. This discrepancy is likely because ACCEL initiates the curriculum with "easy" levels, and gradually introducing complexity via minor mutations, whereas PLR relies on DR, which lacks the fine-grained control over difficulty progression that ACCEL's mutation-based method offers. As a result, while CENIE enhances state-action space coverage for both ACCEL and PLR, it is likely that ACCEL's gradual complexity introduction mechanism capitalizes on this enhancement more effectively.

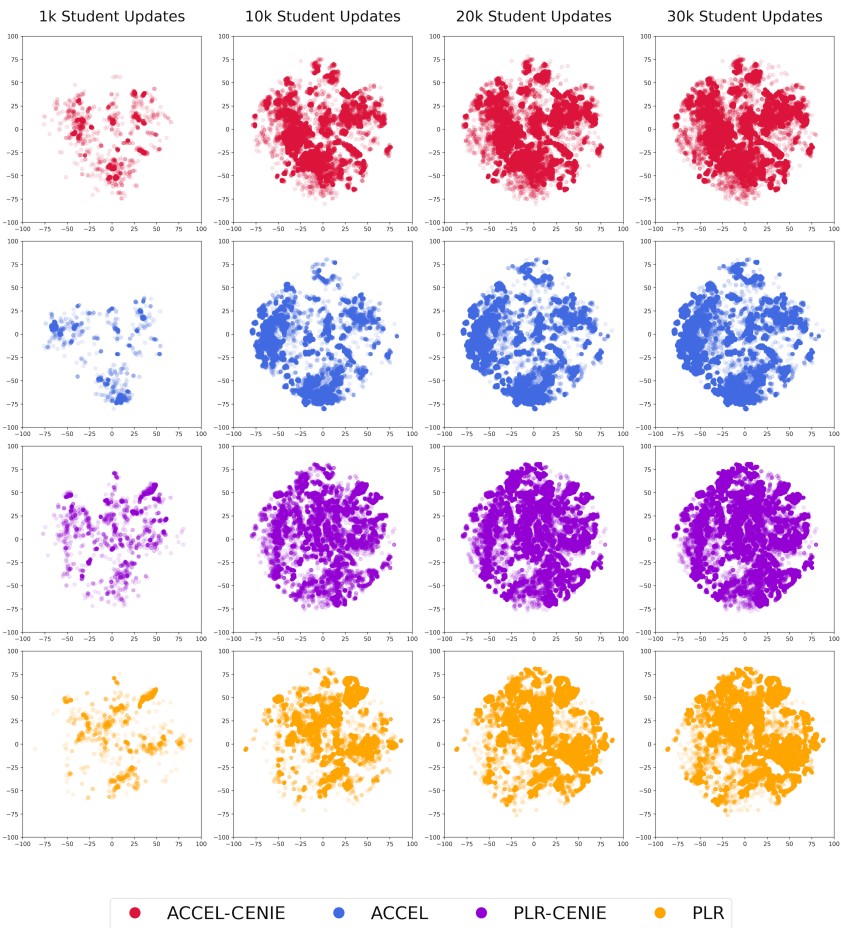

Figure 12: Evolution of the state-action space coverage of ACCEL-CENIE, ACCEL, PLR-CENIE, and PLR for a seed. The checkpoints are 1k, 10k, 20k, and 30k policy updates during the training.

To plot the level difficulty composition of the replayed levels by ACCEL and ACCEL-CENIE in Figure 5 of the main paper, we adapted the difficulty thresholds originally defined in Wang et al. [56]. This is because their thresholds were designed for a smaller 5-D encoding BipedalWalker environment, whereas our setting uses an 8-D encoding, which allows for higher complexity of levels to be generated. Specifically, we introduced an additional threshold for maximum stairs height, as shown in Table 2. A level is classified as Easy if it meets none of the thresholds, and as Moderate, Challenging, Very Challenging, or Extremely Challenging if it meets one, two, three, or four thresholds, respectively.

Note that our Figure 5 differs from Figure 12 in Parker-Holder et al. [37] which shows the difficulty distribution of the levels **generated and added into the buffer**, but not the actual levels selected

Table 2: Environment encoding thresholds for 8D BipedalWalker.

| Stump Height (High) | Pit Gap (High) | Ground Roughness | Stairs Height (High) |
|:---:|:---:|:---:|:---:|
| $\geq 2.4$ | $\geq 6$ | $\geq 4.5$ | $\geq 5$ |

by the teacher for the student to **replay/train on**. Also, their figure is defined for the 5D encoding setting. On that note, this also demonstrates that CENIE remedies an inefficiency in the original ACCEL algorithm, where mutation-based generation is capable of producing high complexity levels but are not selected for student training due to solely depending on regret for level prioritization.

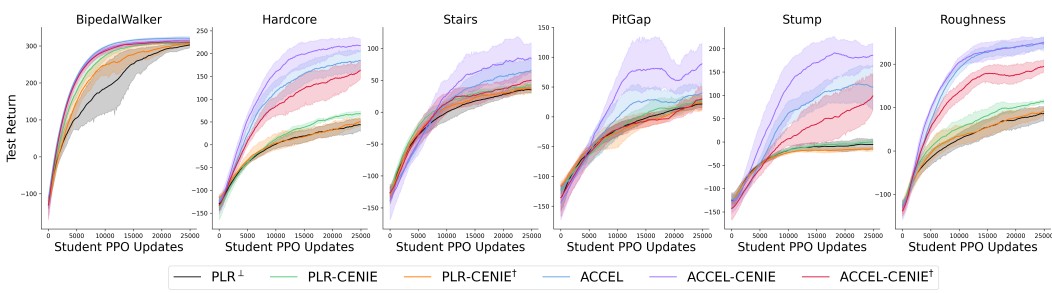

Figure 13: Zero-shot transfer return ablations in BipedalWalker domain. The plot is based on mean and standard error over 5 independent runs.

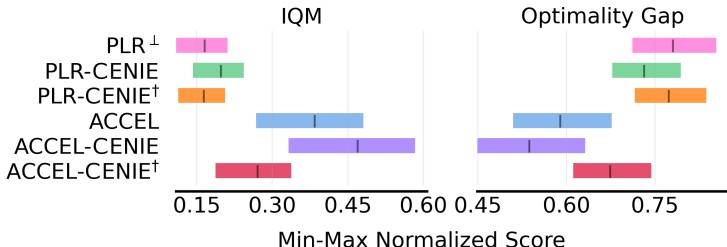

Figure 14: IQM and Optimality Gap ablations in BipedalWalker domain. Results are measured across 5 independent runs.

In addition to the state-action space coverage, we also conduct the ablation study in the BipedalWalker domain. We repeat the same experiment settings as in the Minigrid domain, where both ACCEL-CENIE$^\dagger$ and PLR-CENIE$^\dagger$ utilize only novelty to prioritize replay levels, and ACCEL-CENIE and PLR-CENIE integrate both novelty and regret for prioritization. We assess the algorithm performance with the same evaluations as in the main paper, providing both the transfer performance during training (Figure 13) and IQM and Optimality Gap (Figure 14).

Summarizing the observations from both Figure 13 and Figure 14, we observe that novelty-driven level replay selection exhibits a similar effect as regret on PLR$^\perp$ but is not as effective as regret on ACCEL. PLR-CENIE$^\dagger$ performs on par with the regret metric counterpart (i.e., PLR$^\perp$) while ACCEL-CENIE$^\dagger$ is outperformed by ACCEL and ACCEL-CENIE in this domain. The observations differ from the ablation studies conducted in the Minigrid domain. This discrepancy is possibly due to the greater importance of exploration in Minigrid, which features a sparse reward setting, compared to the dense reward, continuous control domain of BipedalWalker.

### A.3 CarRacing Domain

To monitor the evolution of the students' transfer performance, we evaluate the students every 100 PPO updates on four racing tracks throughout the training period and plot the results in Figure 15. PLR-CENIE outperforms both its predecessor, PLR$^\perp$, and the state-of-the-art algorithm, DIPLR, in the CarRacing domain.

Since both DIPLR and PLR-CENIE achieve near-optimal performance on the four testing tracks, we conduct a more extensive and rigorous evaluation by measuring the students' transfer performance on 20 human-designed F1 racing tracks from Jiang et al. [27]. We also include the ablation model, PLR-CENIE† which uses novelty alone to prioritize replay levels, in our evaluation. The detailed results of each algorithm are listed in Table 3. For better visualization and straightforward comparison, we plotted the IQM and Optimality Gap performances in Figure 16.

Table 3: Test returns of each method on all the CarRacing F1 benchmarks. Results are measured across 5 runs at 2.75k PPO updates and 50 trials per track. Bold indicates being within one standard error of the best mean. Observe that PLR-CENIE consistently outperforms the other algorithms or matches the best-performing algorithm. PLR-CENIE† is the ablation model.

| Track | DR | Minimax | PAIRED | DIPLR | PLR$^\perp$ | PLR-CENIE | PLR-CENIE† |
|---|---|---|---|---|---|---|---|
| Australia | 304±133 | 107±97 | 224±173 | **715±50** | 574±69 | **745±32** | 616±45 |
| Austria | 299±118 | 152±106 | 159±160 | **587±49** | 458±44 | **566±38** | 496±46 |
| Bahrain | 208±136 | 44±101 | 118±159 | **514±48** | 377±75 | **537±58** | 453±38 |
| Belgium | 225±104 | 131±87 | 110±100 | 440±31 | 362±36 | **500±41** | 436±35 |
| Brazil | 192±106 | 57±61 | 147±124 | **451±39** | 368±42 | **485±27** | 312±40 |
| China | -35±57 | -29±80 | -71±63 | 93±102 | -23±28 | **278±100** | **281±52** |
| France | 124±111 | 48±129 | 8±126 | 487±75 | 311±98 | **564±65** | 435±97 |
| Germany | 172±105 | 94±100 | 2±97 | **477±59** | 358±35 | **512±80** | **500±82** |
| Hungary | 319±155 | 133±113 | 139±161 | **686±50** | 597±72 | **678±40** | 604±70 |
| Italy | 267±114 | 204±89 | 198±135 | 676±30 | 559±63 | **708±26** | 588±34 |
| Malaysia | 142±107 | 39±94 | 51±104 | 404±30 | 265±44 | **469±79** | 338±22 |
| Mexico | 331±199 | 193±123 | 102±169 | **675±24** | 570±76 | **674±51** | 602±57 |
| Monaco | 80±78 | 100±94 | 34±111 | 369±122 | 139±112 | **641±46** | 476±96 |
| Netherlands | 143±109 | 104±95 | 42±77 | **540±34** | 400±61 | **558±59** | 403±84 |
| Portugal | 174±118 | 39±94 | 88±153 | 412±22 | 353±27 | **495±66** | 394±43 |
| Russia | 343±151 | 118±105 | 204±163 | **609±60** | **644±31** | 594±58 | 550±60 |
| Singapore | 209±108 | 75±93 | 88±153 | **479±78** | 423±51 | **530±48** | 454±55 |
| Spain | 296±133 | 181±110 | 249±157 | **619±39** | 517±41 | **588±43** | 499±39 |
| UK | 303±127 | 187±101 | 194±156 | **558±49** | 443±45 | **562±26** | 506±36 |
| USA | 173±95 | -2±84 | 2±161 | 191±110 | 155±90 | **416±143** | 363±61 |
| Mean | 214±115 | 99±92 | 105±132 | 499±20 | 392±28 | **553±32** | 465±42 |

From both Table 3 and Figure 16, we observe that the ablation model, PLR-CENIE†, outperforms PLR$^\perp$ by a significant margin, indicating that novelty is more important for level replay prioritization than the regret metric in PLR$^\perp$ for the CarRacing domain. Moreover, PLR-CENIE surpasses DIPLR and achieves state-of-the-art transfer performance in the CarRacing domain by effectively combining the strength of both novelty and regret.

## B  Extended Related Work

**Curiosity-driven Approaches in RL**   CENIE and curiosity-driven RL [47, 50] share a conceptual similarity in leveraging novelty or unfamiliarity to guide learning. However, they differ significantly in their application and theoretical foundations. Curiosity-driven learning seeks to quantify "curiosity" as an intrinsic reward for the agent such that it learns to prioritize the exploration of interesting experiences within a static environment [38], or across a set of predefined tasks [18]. In contrast, CENIE is an autocurricula approach that focuses on curating environments interesting or useful for the agent's learning, shaping the learning curriculum itself rather than the exploration reward signal. This distinction is analogous to the difference between Prioritized Experience Replay [46] in traditional RL and Prioritized Level Replay [28] in UED. The former is an "inner-loop" method prioritizing past experiences for training, while the latter is an "outer-loop" method using past experiences to inform the collection/generation of future experiences. Similarly, curiosity-driven learning prioritizes novel experiences for policy updates, whereas CENIE focuses on generating and curating levels that induce these novel experiences. This fundamental difference in purposes makes theoretical and empirical

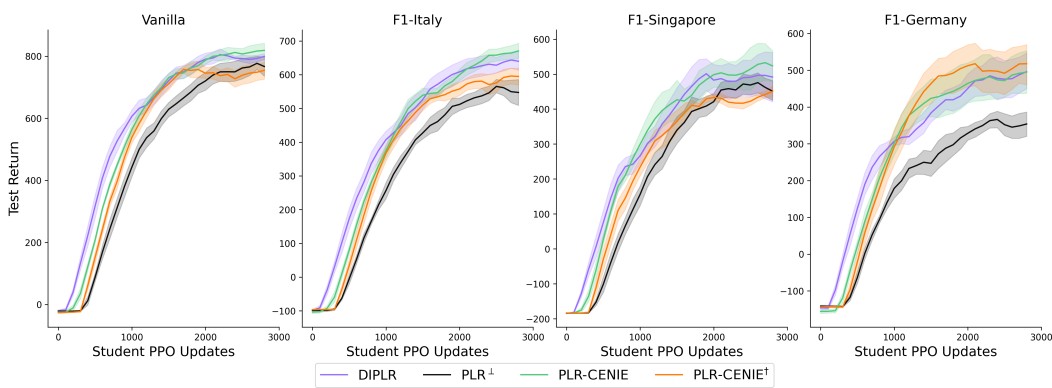

Figure 15: Complete training process of each algorithm on four CarRacing test environments. Plots show mean and standard error over 5 independent runs, with an evaluation interval of 100 PPO updates.

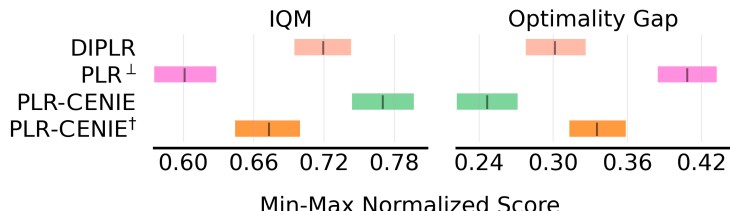

Figure 16: IQM and Optimality Gap ablations on the full CarRacing benchmark (20 F1 tracks). Results are measured across 5 independent runs after 2.75k PPO updates.

comparisons between curiosity-driven approaches and CENIE less direct. However, many of the previous works in curiosity-driven RL provide inspiration for the CENIE framework. Specifically, curiosity-driven RL methods often seek to represent the "visitation counts" of state-action to shape the intrinsic reward. The use of GMMs to model state-action space coverage in CENIE is motivated by successes in curiosity-driven RL approaches which have tackled the counting problem using density models. Notably, density models have been flexibly used to model state-action visitations for both large discrete state-action spaces (via pseudo-counts [8, 36]) and continuous state-action spaces [62, 61].

**Automatic Curriculum Learning**    UED is related to *Automated Curriculum Learning* (ACL;[42]), which emcompasses a family of mechanisms that automatically adapt the distribution of training data by selecting learning situations tailored to the capabilities of DRL agents. Many ACL methods prioritize sampling of environment instances where the agent achieves high *learning progress* (LP). A particular relevant method in this space is *ALP-GMM*, introduced by Portelas et al. [40]. ALP-GMM operates by periodically fitting a Gaussian Mixture Model to a dataset of previously sampled environment parameters, each associated with an Absolute Learning Progress (ALP) score. The approach employs an EXP4 [6] bandit algorithm to select Gaussians as arms, with each Gaussian's utility defined by its ALP score. ALP-GMM's approach to fitting multiple GMMs using different number of Gaussian components and keeping the best one inspired our GMM approach. However, they evaluate the GMM's quality using the Akaike's Information Criterion [15] (AIC). AIC introduces a penalty for the number of parameters in the model (which increases with the number of Gaussian components and dimensions of the data). This penalizes GMMs with more components, which may not be ideal for accurately modeling well-separated clusters in the state-action space which is crucial for identifying sparse regions and estimating novelty. To address this, our work uses the silhouette score [45] instead, which better evaluates clustering quality by considering both intra-cluster cohesion and inter-cluster separation, making it better suited for modeling novelty in state-action spaces. Additionally, ALP-GMM uses GMMs to sample environment parameters that are likely to yield high

ALP scores, aligning task difficulty with the agent's progress. This approach contrasts with how GMMs are used in the CENIE framework, where the goal is to model the novelty of an environment based on state-action coverage, independent of specific environment parameters. ACL methods like ALP-GMM generally assume a predefined target task distribution. This differs from the UED framework which only requires only an underspecified task space, i.e. $\theta$ in the UPOMDP formalism. UED seeks to directly maximize the student's robustness over any possible environments, even those that are out-of-distribution from training. Similarly, CENIE provides a general approach for quantifying novelty using state-action coverage, without relying on any predefined task distribution. Due to this generality, we believe the CENIE framework holds significant potential for crossover applications in the ACL domain, providing a robust method for assessing and prioritizing environment novelty to enhance curriculum learning.

**Open-endedness and Novelty Search in Evolutionary Computation** Long-running UED processes in expansive UPDOMPs closely resemble continual learning in open-ended domains. As such, UED is fundamentally connected to the fields of *open-endedness* [51] and evolutionary computation. When the task space allows for unbounded complexity, autocurricula methods such as UED offer promising pathways to open-endedness by co-evolving an adaptive, infinite set of tasks for the agent. Traditionally, learning without extrinsic rewards or fitness functions has been studied in evolutionary computation where it is referred to as 'novelty search' [30, 31]. In novelty search, the novelty of an agent's behavior is typically quantified by measuring the distance between a user-defined feature or behavioral descriptor and its nearest neighbor in the population. Consistent with our findings, the open-ended learning literature has long recognized that high-performing solutions often emerge not through fitness optimization alone but through novelty-driven exploration. Despite these parallels, novelty search in environment design remains underdeveloped. Early work such as POET [56] and its successor [57] in open-ended RL have started drawing connections, linking environment design with principles of open-ended exploration. However, these approaches rely on a population of agents and distance-based novelty measures that lack curriculum-awareness; they do not adapt to the specific experiences induced by the curriculum nor improve the agent's sample efficiency in reducing uncertainty across the state-action space. More recent work by Zhang et al. [60] proposed to leverage foundation models to quantify human notions of "interestingness" (e.g. tasks that are both novel and worthwhile) in order to narrow the environment search space. It is unclear how to combine the insights from Zhang et al. [60] and this paper. Integrating these insights with our work presents an intriguing challenge. On one hand, CENIE provides a principled, general approach to quantifying novelty through state-action coverage, circumventing the need for subjective evaluations of "interestingness" using foundation models. On the other hand, Zhang et al. [60] points out critical pitfalls in novelty search, such as the potential for agents to exploit novelty measures, generating superficial variations that fail to yield genuinely meaningful insights. This highlights numerous exciting research directions for aligning novelty search with the concept of "interestingness," potentially combining the strengths of principled coverage-based novelty measures with more nuanced assessments of task value.

## C    Future Work and Limitations

In this paper, we demonstrated the application of GMMs to quantify the novelty of environments generated under the UED paradigm. We then validated the effectiveness of this novelty metric in prioritizing levels. Nevertheless, our work has some limitations. First, while we demonstrated the utility of the CENIE framework for novelty quantification and level prioritization, we did not explore its potential for directly generating novel environments. We anticipate that with creative manipulations, the GMM likelihood scores could directly inform level generation, either through a principled level generator (as in PAIRED) or by guiding mutations (as in ACCEL). This approach may lead to a more sample-efficient generation process, reducing the variance inherent in random generation.

Second, we did not experiment with alternative weightings between regret and CENIE's novelty in level replay prioritization, as our experiments used a fixed 0.5-0.5 weighting (as in Eq.5). We hypothesize that tuning these weights based on domain characteristics, such as the required level of exploration or reward sparsity, could improve performance. Additionally, employing dynamic weighting schemes, such as linearly decaying weight adjustments or adaptive strategies based on the agent's learning progress, may further enhance curriculum optimization.

Third, GMM-based clustering may encounter challenges due to the curse of dimensionality in high-dimensional state-action spaces. While our current CENIE-augmented algorithms demonstrated significant improvements, future work could explore dimensionality reduction techniques, such as Principal Component Analysis (PCA; [25]) or t-distributed Stochastic Neighbor Embedding (t-SNE; [54]), to improve coverage representation in higher-dimensional settings. However, even with dimensionality reduction, such representations may still struggle in environments where the observations contain exogenous information irrelevant to the agent's control. Specifically, although our experiments showed strong empirical gains in simplified environments, the current approach is vulnerable to the noisy TV problem [19], where novelty-driven level prioritization may focus on unpredictable noise elements of the environment, rather than beneficial learning experiences. This limitation highlights the importance of balancing level prioritization between novelty and regret to ensure the agent focuses on genuinely novel environments rich in learning potential.

Furthermore, effective state representation is crucial for the CENIE framework. The CENIE framework is not restricted to raw state inputs; it can operate on indirect encodings, such as latent-space representations obtained from a generative model of the environment. This approach would allow CENIE to capture action-relevant information and necessary temporal dependencies between states, providing a more focused basis for novelty estimation. We expect that density-based novelty estimation could improve further by using latent representations from more expressive generative models, such as Variational Autoencoders (VAEs) [29], which can capture richer, more informative structures in the state space.

Finally, while we used GMMs for environment novelty quantification, the CENIE framework is not limited to this model, as mentioned in the main body of this paper. GMMs may face limitations in capturing more complex distributions in real-world settings, and our choice of GMMs was primarily intended to illustrate the empirical benefits of quantifying novelty using state-action coverage in simpler environment settings. It is important to point out that fitting multiple GMM on the updated state-action coverage distribution and selecting the best one every rollout can incur additional computational costs. For future work aiming to replicate our approach, exploring periodic refitting (similar to the strategy used in ALP-GMM) could be worthwhile, as it may achieve comparable effectiveness while significantly reducing computational demands. Future work could also investigate more advanced density models, such as Variational Gaussian Mixture Models [11], Deep Gaussian Mixture Models [55], or Normalizing Flow Models [44]. Additionally, there may be alternative approaches beyond density models for representing state-action coverage that could further enhance CENIE's effectiveness. We believe there are many promising directions for the CENIE framework, and we leave these potential extensions to future work.

## D   Implementation Details

In this section, we provide the details about the experiments and implementations, including domain properties and additional information about CENIE and the baseline algorithms. All of our experiments are run with a single V100 GPU or GeForce 3090 GPU, using 10 Intel Xeon E5-2698 v4 CPUs. The baseline algorithms and evaluation environments are implemented using the DCD codebase provided by Jiang et al. [27], Parker-Holder et al. [37]. The CENIE framework and our current evaluations build upon and significantly extend a preliminary version of our work [52], where the framework was initially named "GENIE." We have since enhanced the framework and opted to rename it to CENIE, following the release of a similarly-named, related work by Bruce et al. [17], which appeared around the same time. This change was made to distinguish our contributions clearly and avoid confusion within the research community.

### D.1   Fitting Gaussian Mixture Models

In this section, we provide more details about the GMM fitting process that was absent from the main body. Given an initial buffer containing past state-action pairs, $\Gamma$, and a selected number of Gaussians, $K$, we first use the *k-means++* algorithm to perform a fast and efficient initialization of the GMM parameters [12, 4], $\lambda_\Gamma = \{(\alpha_1, \mu_1, \Sigma_1), ..., (\alpha_K, \mu_K, \Sigma_K)\}$. We then optimize $\lambda_\Gamma$ using the Expectation Maximization (EM) algorithm [22, 43]. The EM algorithm uses the initial values $\lambda_\Gamma$ to estimate a new $\lambda'_\Gamma$ such that $P(X|\lambda'_\Gamma) > P(X|\lambda_\Gamma)$. This process is repeated iteratively until some convergence threshold is fulfilled. Each iteration of the EM algorithm can be separated into the

E-step and M-step. In E-step, the posterior probability for each component $i$ generating the sample point $x_t$ is denoted by $w_{t,i}$,

$$w_{t,i} = P(i|x_t) = \frac{\alpha_i \mathcal{N}(x_t|\mu_i, \sigma_i)}{\sum_i^K \alpha_i \mathcal{N}(x_t|\mu_i, \sigma_i)}$$

where $t = 1, 2, ..., N$ and $i = 1, 2, ..., K$. M-step computes the maximum likelihood estimation (MLE) using $w_{t,i}$ following re-estimation formulas which are derived from the partial derivatives of the log-likelihood functions and guarantee a monotonic increase in the model's likelihood value.

$$\alpha_i = \frac{1}{N} \sum_{i=1}^{N} w_{t,i}, \quad \mu_i = \frac{\sum_{i=1}^{N} w_{t,i}}{\sum_{i=1}^{N} w_{t,i}} x_t$$

$$\sigma_i = \frac{\sum_{i=1}^{N} w_{t,i}}{\sum_{i=1}^{N} w_{t,i}} (x_t - \mu_i)(x_t - \mu_i)^T$$

We iteratively apply the E-step and M-step until the parameters converge, i.e., $||\lambda_\Gamma' - \lambda_\Gamma|| < \epsilon$, where $\epsilon$ is a small threshold.

As mentioned in the main paper, we deliberately employ a finite window for $\Gamma$ to account for the effects of catastrophic forgetting. This allows levels with state-action pairs encountered in the past but subsequently forgotten by the agent's policy to regain novelty and be included back in the agent's training curriculum. Furthermore, to ensure effectiveness in clustering the state-action space, we utilized a semi-online GMM model that is able to adapt its number of Gaussians, i.e. $K$, to that of the highest silhouette score.

We use the `PyCave` [13] Python library to fit the GMM using GPU acceleration, which also provides an efficient abstraction for the Expectation-Maximization (EM) algorithm. We use the `PyTorch Adapt` [35] Python library to calculate the silhouette scores. The hyperparameters for fitting the GMM for all domains are shown in Table 7.

### D.2 CENIE-Augmented Algorithms

Besides the algorithm for ACCEL-CENIE shown in the main paper under Algorithm 1, we also provide the algorithm for PLR-CENIE here under Algorithm 2.

### D.3 Minigrid Domain

In the Minigrid domain, the teacher creates maze instances consisting of a $15 \times 15$ grid, where each empty tile can be occupied by the agent, the goal, an obstacle (i.e. block), or an empty space that can navigate through. The student is aware of its orientation and is limited by partial observability, i.e. it only has a $5 \times 5$ view in front of it. The student agent can only move forward and turn left/right, and will stay in place if it hits an obstacle. The student agent is implemented based on PPO [49] with an LSTM-based recurrent network structure to deal with partial observability. We use the LSTM hidden states as representations within our GMM, allowing the density model to capture temporal dependencies between states. The student agent receives a reward upon reaching the goal, where $H$ is the episode length and $H_{max}$ is the maximum length (set to 250 at training) for an episode. The agent receives a reward of $r = 1 - (num_{step}/H_{max})$ when it reaches the goal position and 0 if it fails to reach the goal. The collection of states in this domain depicts the scenarios the agent needs to navigate through.

### D.4 BipedalWalker Domain

In BipedalWalker, the teacher agent generates new levels by specifying the values of the eight environment parameters (e.g., ground roughness, number of stair steps, pit gap width, etc). As for the student agent, it needs to determine the torques applied on its joints and is constrained by partial observability where it only knows its horizontal speed, vertical speed, angular speed, positions of joints, etc. The student agent receives positive rewards as it walks towards the goal position and will receive a large negative penalty if it falls down. The BipedalWalker domain is modified on top of the BipedalWalkerHardcore environment from OpenAI Gym, introduced by [56] and improved by [41, 37]. The student agent receives a 24-dimensional proprioceptive state corresponding to inputs

**Algorithm 2** PLR-CENIE

**Input**: Level buffer size $N$, Component range $[K_{\min}, K_{\max}]$, FIFO window size $\mathcal{W}$, random level generator $\mathcal{G}$

**Initialize**: Student policy $\pi_\eta$, level buffer $\mathcal{B}$, state-action buffer $\Gamma$, GMM parameters $\lambda_\Gamma$

1: Generate $N$ initial levels by $\mathcal{G}$ to populate $\mathcal{B}$
2: Collect $\pi_\eta$'s trajectories on each level in $\mathcal{B}$ and fill up $\Gamma$
3: **while** not converged **do**
4:     Sample replay decision, $\epsilon \sim U[0, 1]$
5:     **if** $\epsilon \geq 0.5$ **then**
6:         Generate a new level $l_\theta$ by $\mathcal{G}$
7:         Collect trajectories $\tau$ on $l_\theta$, with stop-gradient $\eta_\perp$
8:         Compute novelty score for $l_\theta$ using $\lambda_\Gamma$ (Eq.3 and Eq.4)
9:         Compute regret score for $l_\theta$ (Eq.1 and Eq.4)
10:        Update $\mathcal{B}$ with $l_\theta$ if $P_{replay}(l_\theta)$ is greater than that of any levels in $\mathcal{B}$ (Eq.5)
11:     **else**
12:         Sample a replay level $l_\theta \sim \mathcal{B}$ according to $P_{replay}$
13:         Collect trajectories $\tau$ on $l_\theta$
14:         Update $\pi_\eta$ with rewards $R(\tau)$
15:         Compute novelty score for $l_\theta$ using $\lambda_\Gamma$ (Eq.3 and Eq.4)
16:         Compute regret score for $l_\theta$ (Eq.1 and Eq.4)
17:         Update $P_{replay}$ with novelty and regret scores
18:         Update $\Gamma$ with $\tau$ and resize to $\mathcal{W}$
19:         **for** $k$ in range $K_{\min}$ to $K_{\max}$ **do**
20:             Fit a $GMM_k$ with $k$ components on $\Gamma$ and compute its silhouette score
21:         **end for**
22:         Select GMM parameters with the highest silhouette score to replace $\lambda_\Gamma$
23:         Collect trajectories $\tau$ on $l_\theta$, with stop-gradient $\eta_\perp$
24:         Update $\mathcal{B}$ with $l_\theta$ if $P_{replay}(l_\theta)$ is greater than that of any levels in $\mathcal{B}$ (Eq.5)
25:     **end if**
26: **end while**

from its lidar sensors, angles, and contacts, which also form the state representation for our GMM. The partial observability here means the agent does not have access to its positional coordinates. The environment parameters and their corresponding ranges are shown in Table 4. Note, there will be a singular value to specify Ground Roughness and the Number of Stair Steps, and a min and a max value to define the PitGap Width, Stump Height, and Stair Height, and thus we will have eight environment parameters in total.

Table 4: Environment parameters and their ranges in the BipedalWalker domain. To define PitGap, StumpHeight, and StairHeight, we need a min and a max value. Hence, there are a total of eight parameters.

| Parameter | Roughness | Num of Stair Steps | PitGap Width | Stump Height | Stair Height |
|---|---|---|---|---|---|
| Range | [0,10] | [1,9] | [0,10] | [0,5] | [0,5] |

### D.5 CarRacing Domain

The CarRacing domain was introduced and customized by [27]. In CarRacing, the teacher creates tracks by using Bézier curves by 12 control points within a fixed radius of the center of the playfield. A track consists of a sequence of $L$ polygons and $L$ is fixed on the training tracks and varies on different testing tracks. While driving on the tracks, the student receives a reward equal to $1000/L$. The student additionally receives a reward of $-0.1$ at each time step. The student observes an RBG image of size (96x96x3), where (96x96) is the (height x width) of the observed image and 3 is the number of RGB channels. Consistent with previous UED literature, we employ a CNN model to preprocess the raw image observations. The CNN extracts high-level features from the raw images, reducing their dimensionality and capturing important spatial patterns that are crucial for

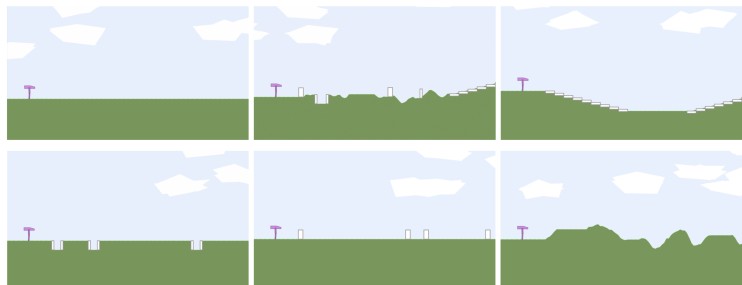

Figure 17: Examples of testing levels in BipedalWalker domain. (a) BipedalWalker, (b) Hardcore, (c) Stair, (d) PitGap, (e) Stump, and (f)Roughness.

understanding the environment's dynamics. By feeding these feature representations into the GMM model instead of the raw image input, we ensure that the density estimation focuses on meaningful aspects of the state space rather than being overwhelmed by the complexity and noise of raw pixel data.

The 20 testing F1 tracks by [27] have various track lengths and different maximum episode steps. As such, different min-max normalization ranges are used for each track to produce the IQM and Optimality Gap plots. We list the 20 test tracks and their corresponding min-max normalization ranges in Table 5.

Table 5: Min-max ranges for different CarRacing F1 tracks that are used for IQM and Optimality Gap plotting.

| Track | Episode Steps | Min-Max Reward Range |
|---|---|---|
| Australia
Austria
Belgium
Italy
Monaco | 1500 | [-150, 850] |
| Brazil
China
France
Germany
Hungary
Netherlands
Russia
Singapore
Spain
UK
USA | 2000 | [-200, 800] |
| Bahrain
Malaysia
Portugal | 2500 | [-250, 750] |
| Mexico | 3000 | [-300, 700] |

# E   More Details On Baseline Algorithms

In this section, we provide more technical details on some of the baseline algorithms used in our experiments, specifically Domain Randomization (DR), Minimax, PAIRED, $PLR^{\perp}$, DIPLR, and ACCEL. We summarize the key differences between the baseline algorithms in Table 6.

The Domain Randomization (DR) teacher uniformly randomizes each dimension in the environment parameter space to generate various environments.

Table 6: Overview of the fundamental UED algorithms and CENIE-augmented algorithms.

| Algorithm | Generation Strategy | Generator Obj | Curation Obj | Setting |
|---|---|---|---|---|
| POET | Mutation | Minimax | MCC | Population |
| PAIRED | RL | Minimax Regret | None | Single-agent |
| PLR$^\perp$ | Random | None | Minimax Regret | Single-agent |
| DIPLR | Random | None | Minimax Regret + Diversity | Single-agent |
| ACCEL | Random + Mutation | Minimax Regret | Minimax Regret | Single-agent |
| PLR-CENIE | Random | None | Minimax Regret + Novelty | Single-agent |
| ACCEL-CENIE | Random + Mutation | Minimax Regret + Novelty | Minimax Regret + Novelty | Single-agent |

The PAIRED teacher estimates regret by leverage two agents: an antagonist agent and a protagonist agent (student). In practice, PAIRED derives regret by taking the antagonist's ($A$) maximum performance and the protagonist's ($P$) average performance over several trajectories, allowing for more accurate approximations. Let $U^\pi(\tau)$ denote the total reward obtained by a trajectory $\tau$ produced by policy $\pi$ on the level $\theta$. Regret is measured in PAIRED via:

$$\text{REGRET}^\theta(\pi_A, \pi_P, \theta) = \max_{\tau^A} U^\theta(\tau^A) - \mathbb{E}_{\tau^P}[U^\theta(\tau^P)]$$

where $\pi^A$ and $\pi^P$ are the antagonist's policy and the protagonist's policy, respectively. PAIRED teacher constantly creates levels that are slightly beyond the ability range of the protagonist and within the ability range of the antagonist such that the regret is maximized. The pseudocode of the PAIRED algorithm is given in Algorithm 3.

---

**Algorithm 3** PAIRED

**Input**:Randomly initialize Protagonist $\pi^P$, Antagonist $\pi^A$, and teacher $\Lambda$
**Initialize**: replay buffers $\mathcal{B}$

1: **while** not converge **do**
2:     Use teacher to generate environment parameters: $\theta \sim \Lambda$. Use $\theta$ to create environments, $l_\theta$
3:     Collect Protagonist trajectory $\tau^P$ in $l_\theta$. Compute Protagonist's average return: $\mathbb{E}^\theta[V(\pi^P)]$
4:     Collect Antagonist trajectory $\tau^A$ in $l_\theta$. Compute Antagonist's average return: $\mathbb{E}^\theta[V(\pi^A)]$
5:     Compute regret: REGRET = $\mathbb{E}^\theta[V(\pi^A)]$ - $\mathbb{E}^\theta[V(\pi^P)]$
6:     Train Protagonist policy $\pi^P$ with RL update and reward = -REGRET
7:     Train Antagonist policy $\pi^A$ with RL update and reward = REGRET
8:     Train teacher policy with RL update and reward = REGRET
9: **end while**

---

However, the PAIRED algorithm faces several drawbacks [34]. Both the antagonist and protagonist policies are constantly updating, making the problem nonstationary. Furthermore, PAIRED suffers from a long-horizon credit assignment problem since the teacher must fully specify an environment before receiving a sparse reward in the form of feedback from the antagonist and protagonist agents. PLR seeks to circumvent this issue through the use of regret for prioritized selection of levels for replay rather than active generation. PLR uses *Positive Value Loss* (PVL), an approximation of regret based on Generalized Advantage Estimation (GAE; [48]):

$$\text{PVL}^\theta(\pi) = \frac{1}{T} \sum_{t=0}^{T} \max \left( \sum_{k=t}^{T} (\gamma\lambda)^{k-t} \delta_k^\theta, 0 \right),$$

where $\gamma$, $\lambda$ and $T$ are the MDP discount factor, GAE discount factor and MDP horizon, respectively. $\delta_k^\theta$ is the TD-error at time step $k$ for $\theta$. However, the use of PVL may introduce bias due to the bootstrapped value target. An alternate heuristic score function is *Maximum Monte Carlo* (MaxMC),

which replaces the bootstrapped value target with the highest return observed on the level during training. By using this maximal return, the regret estimates become independent of the agent's current policy:

$$\text{MAXMC}^{\theta}(\pi) = \frac{1}{T} \sum_{t=0}^{T} \left( R_{\max}^{\theta} - U(\tau^{\pi}) \right),$$

where $R_{\max}^{\theta}$ is the maximal return of $\pi$ on $\theta$. We primarily focus on PVL because the original implementations of ACCEL and PLR$^{\perp}$ in Jiang et al. [27], Parker-Holder et al. [37] found better success with the PVL scoring function for the experiments domains, i.e. Minigrid, BipedalWalker, and CarRacing, used in this paper. Future research could explore the potential of using the MaxMC scoring function to see if it yields different outcomes when combining ACCEL and PLR$^{\perp}$ with CENIE. The Diversity Induced Prioritized Level Replay (DIPLR [33]) algorithm extends PLR$^{\perp}$ by prioritizing level replay based on both regret and diversity. Here, diversity is quantified using the Wasserstein distance between the agent's trajectories across levels in the replay buffer. The limitations of DIPLR are highlighted in the main body (see Section 3). The pseudocode of DIPLR is provided in Algorithm 4.

---

**Algorithm 4** DIPLR

---

**Input**: Level buffer size $N$, level generator $\mathcal{G}$
**Initialize**: student policy $\pi_{\eta}$, level buffer $L$, trajectory buffer $\Gamma$

 1: Generate $N$ initial levels by $\mathcal{G}$ to populate $L$
 2: Collect trajectories on each replay level in $L$ and fill up $\Gamma$
 3: **while** not converged **do**
 4:     Sample replay-decision, $\epsilon \sim U[0, 1]$
 5:     **if** $\epsilon \geq 0.5$ **then**
 6:         Generate a new level $l_{\theta_i}$ by $\mathcal{G}$
 7:         Collect trajectories $\tau_i$ on $l_{\theta_i}$, with stop-gradient $\eta_{\perp}$
 8:         Compute the regret, staleness and distance for $l_{\theta_i}$
 9:     **else**
10:         Sample a replay level $l_{\theta_j} \in L$ according to $P_{replay}$
11:         Collect trajectories $\tau_j$ on $l_{\theta_j}$ and update $\pi_{\eta}$ with rewards $R(\tau_j)$
12:         Compute the regret, staleness and distance for $l_j$
13:     **end if**
14:     Flush $\Gamma$ and collect trajectories on all replay levels to fill up $\Gamma$
15:     Update regret, staleness, and distance for $l_{\theta_i}$ or $l_{\theta_j}$
16:     Update $L$ with new level $l_{\theta_i}$ if its replay probability is greater than any levels in $L$
17:     Update replay probability $P_{replay}$
18: **end while**

---

Finally, the state-of-the-art UED algorithm, ACCEL, improves PLR$^{\perp}$ by replacing its random level generation with an editor that mutates previously curated levels to gradually introduce complexity into the curriculum. ACCEL makes the key assumption that regret varies smoothly with the environment parameters $\theta$, such that the regret of a level is close to the regret of others within a small edit distance. If this is the case, then small edits to a single high-regret level should lead to the discovery of entire batches of high-regret levels – which could be an otherwise challenging task in high-dimensional design spaces. An intriguing area for future exploration is the interaction between ACCEL's editing mechanism and the novelty-driven level prioritization introduced through CENIE. Specifically, it is worth investigating whether the editing mechanism does synergize with CENIE to produce levels that simultaneously maximize **both novelty and regret**, further enhancing the diversity and effectiveness of the generated curriculum.

## F   Hyperparameters

In this section, we provide the hyperparameters we used for both CENIE-augmented and baseline algorithms in our experiments. We employ the same set of CENIE parameters for both ACCEL-CENIE and PLR-CENIE. We provide all the parameters for our implementations in Table 7.

Table 7: Hyperparameters used for training PLR-CENIE and ACCEL-CENIE in Minigrid, Bipedal-Walker and CarRacing domains. Note that the we inherit the original PLR$^{\perp}$ and ACCEL hyperparameters and only adjust the CENIE hyperparameters.

| Parameter | Minigrid | BipedalWalker | CarRacing |
|---|---|---|---|
| ***PPO*** | | | |
| $\gamma$ | 0.995 | 0.99 | 0.99 |
| $\lambda_{GAE}$ | 0.95 | 0.9 | 0.9 |
| PPO rollout length | 256 | 2048 | 125 |
| PPO epochs | 5 | 5 | 8 |
| PPO minibatches per epoch | 1 | 32 | 4 |
| PPO clip range | 0.2 | 0.2 | 0.2 |
| PPO number of workers | 32 | 16 | 16 |
| Adam learning rate | 1e-4 | 3e-4 | 3e-4 |
| Adam $\epsilon$ | 1e-5 | 1e-5 | 1e-5 |
| PPO max gradient norm | 0.5 | 0.5 | 0.5 |
| PPO value clipping | yes | no | no |
| return normalization | no | yes | yes |
| value loss coefficient | 0.5 | 0.5 | 0.5 |
| student entropy coefficient | 0.0 | 1e-3 | 0.0 |
| | | | |
| ***PLR$^{\perp}$*** | | | |
| Scoring function | positive value loss | positive value loss | positive value loss |
| Replay rate, $p$ | 0.5 | 0.5 | 0.5 |
| Buffer size, $K$ | 4000 | 1000 | 8000 |
| | | | |
| ***ACCEL*** | | | |
| Edit rate, $q$ | 1.0 | 1.0 | N/A |
| Replay rate, $p$ | 0.8 | 0.9 | N/A |
| Buffer size, $K$ | 4000 | 1000 | N/A |
| Scoring function | positive value loss | positive value loss | N/A |
| Edit method | random | random | N/A |
| Number of edits | 5 | 3 | N/A |
| Levels edited | batch | batch | N/A |
| Prioritization, $\beta$ | 0.3 | 0.1 | N/A |
| Staleness coefficient, $\rho$ | 0.5 | 0.5 | N/A |
| | | | |
| ***CENIE*** | | | |
| Initialization strategy | k-means++ | k-means++ | k-means++ |
| Convergence threshold, $\epsilon$ | 0.001 | 0.001 | 0.001 |
| GMM components | [6,15] | [6,15] | [6,15] |
| Covariance regularization | 1e-2 | 1e-6 | 1e-1 |
| Window size (no. of levels) | 32 | 32 | 32 |
| Novelty coefficient | 0.5 | 0.5 | 0.5 |

