# OpenReview forum: "Improving Environment Novelty Quantification for Effective Unsupervised Environment Design"
_NeurIPS.cc/2024/Conference — NeurIPS 2024 oral_

### Official Review · Reviewer_jT1H · 2024-06-27

**Soundness:** 3
**Presentation:** 4
**Contribution:** 3
**Rating:** 7
**Confidence:** 4

**Summary:**

After rebuttal

I have upped my score to 7. I think this paper is good, as it makes a small, easy to implement and simple to understand change to existing UED methods, and it delivers improved empirical results.


-----


This paper uses a GMM-based method to quantity novelty of levels in UED. It uses the state-action distribution (unordered, i.e., not trajectory distribution) induced by the agent on a level and fits a GMM to the data from previously sampled levels. A new level's novelty can be determined by computing the likelihood of its induced state-action distribution under the fitted GMM.

**Strengths:**

- The idea of using state-action distributions to represent a level is not new, but it makes sense.
- Using a fitted model and computing the likelihood using this also makes a lot of sense, compared to computing a pairwise novelty score between two levels.
- Having the ability to have different numbers of GMM kernels is an interesting idea, and seems to provide a good way of allowing complexity to be dynamically altered.
- Results demonstrate that the new method performs better than PLR/ACCEL, and the change in code is small.

**Weaknesses:**

- While the focus on unordered transition tuples is mentioned as a benefit, I would think that there are certain environments where the temporal nature of trajectories are important. Could you comment on this please?
- Minor
	- Line 88, for minimax one also often cites [1]
	- The $\tau$'s beneath the max/expectation in equation (1) don't have the A and P superscripts.

[1] Pinto, Lerrel, et al. "Robust adversarial reinforcement learning." _International Conference on Machine Learning_. PMLR, 2017.

**Questions:**

- Algorithm 1, please indicate what the blue text means (I assume changes to ACCEL but being clear on this would be helpful.)
- Please provide aggregate results averaged over all of the minigrid eval. levels. It is hard at a glance to see how the algorithms compare.
- Do the GENIE generated levels actually look more diverse than those generated by e.g. ACCEL? Please show a sampling of levels.
- In table 1, PLR-GENIE has the most state-action coverage, but performs much worse than the ACCEL-based methods. This seems to contradict the claim that the increased diversity is causing better performance. Could you explain this please?
- There may be some confusion between your GENIE and [1]
- For car racing you do use images, is the image simply flattened? Could this have problems with e.g. not being translationally invariant?

[1] Bruce, Jake, et al. "Genie: Generative interactive environments." _Forty-first International Conference on Machine Learning_. 2024.

**Limitations:**

- I think putting limitations in the main paper would be better. Figure 5 feels not super necessary and can be moved to the appendix if space is an issue.
- Some other limitations I can think of, please comment/explain
	- Choosing the range of $K$ (number of GMM) kernels can be challenging?
	- Do I understand correctly that you just concatenate the observation and actions to form $x$, which is then used to fit the GMM? How can this scale to much larger observations?

---

> ### Author Rebuttal · Authors · 2024-08-06
>
> We appreciate the reviewer's careful attention to detail and the positive feedback provided. The following clarifications will hopefully address your concerns and strengthen the case for our paper:
>
> ### Weaknesses
>
> **Q: While the focus on unordered transition tuples is mentioned as a benefit, I would think that there are certain environments where the temporal nature of trajectories are important. Could you comment on this please?**
>
> A: Revieiwer kpCs brought up the same point and we have answered it above (see question 1 in discussion with Revieiwer kpCs). TL;DR: agree and GENIE's design choice on prioritising individual transition tuples is flexible to accommodate non-Markovian states.
>
> **Q: Minor: 1. Line 88, for minimax one also often cites [1]; 2. The $\tau$'s beneath the max/expectation in equation (1) don't have the A and P superscripts.**
>
> A: Appreciate the sharp eye, we have revised this.
>
> ### Questions
>
> **Q: Algorithm 1, please indicate what the blue text means (I assume changes to ACCEL but being clear on this would be helpful.)**
>
> A: Thank you, we have made this clearer in the revised manuscript.
>
> **Q: Please provide aggregate results averaged over all of the minigrid eval. levels. It is hard at a glance to see how the algorithms compare.**
>
> A: Figure 4a is what you are looking for. The IQM and Optimality Gap are metrics introduced by the rliable library (Agarwal et al., 2021) used for fair aggregation of performances across different tasks (levels) and comparing the performance of algorithms.
>
> **Q: Do the GENIE generated levels actually look more diverse than those generated by e.g. ACCEL? Please show a sampling of levels.**
>
> A: Answered in our global rebuttal. The sections "GENIE Introduces Level Complexity", "Low Regret but High Novelty Levels Provide Interesting Experiences" and their accompanying plots visually demonstrates the diversity of the levels generated by GENIE.
>
>
> **Q: In table 1, PLR-GENIE has the most state-action coverage, but performs much worse than the ACCEL-based methods. This seems to contradict the claim that the increased diversity is causing better performance. Could you explain this please?**
>
> A: It is important to note that there are fundamental differences in the curriculum generation mechanisms of ACCEL and PLR outside of GENIE's control. ACCEL mechanism initiates the curriculum with "easy" levels (e.g. a Minigrid with no walls and only the goal) and leverages minor edits (mutation) to gradually introduce complexity to the levels. In contrast, PLR relies on domain randomization (DR) to generate new levels. DR lacks the fine-grained control over the difficulty progression that ACCEL's mutation-based method offers. As a result, even though GENIE exposes the PLR agent to a wider coverage of state-action pairs, the PLR teacher does not present these experiences to the student in an order that facilitates optimal learning. The inherent difference in curriculum generation mechanism between the two algorithms (i.e. ACCEL and PLR) admits a significant difference in performance from the get-go that cannot be recovered by GENIE. To summarize, GENIE enhances the state-action space coverage for both ACCEL and PLR, but ACCEL's gradual curriculum complexity introduction mechanism simply capitalizes on that better.
>
> **Q: There may be some confusion between your GENIE and [1]**
>
> A: Acknowledged in global rebuttal.
>
> **Q: For car racing you do use images, is the image simply flattened? Could this have problems with e.g. not being translationally invariant?**
>
> A: All algorithms incorporate a CNN model over the images, in accordance with previous UED literature. We thank you for bringing up this point and we have included a small section in the appendix to make this clear.
>
> **Q: Choosing the range of $K$ (number of GMM) kernels can be challenging?**
>
> A: Actually, $K$ does not need to be constricted to a fixed range and can be adapted online. Metrics like silhouette score (that we used) or other metrics such as AIC/BIC provide a score for the fit of the GMM and Bayesian methods can be applied to search for the $K$ (not bounded to a range) that fulfils a desired threshold of the metric. The reason why we used a fixed range for $K$ in GENIE is simply because we found that a range of 6-15 kernels already provided significant improvements to the GENIE-augmented algorithms and did not necessitate further optimization.
>
> **Q: Do I understand correctly that you just concatenate the observation and actions to form $x$, which is then used to fit the GMM? How can this scale to much larger observations?**
>
> A: Yes, your understanding is correct. Regarding the dimensionality issue, it is an issue which can be easily remedied. Dimensionality reduction techniques such as Principal Component Analysis (PCA) or learned autoencoders can be employed to scale down large observation spaces without significant loss of information. This is a direction that is also mentioned in the "Future Work and Limitations" section of the appendix of the main paper. High-dimensional state spaces present issues for deep learning methods in general, and remedies used by the policy algorithm to scale down the state space can be applied in parallel for the fitting of GMMs in GENIE.

---

> > ### Comment · Reviewer_jT1H · 2024-08-07
> > **Discussion**
> >
> > Thank you for your detailed response!
> >
> > A few more follow ups
> >
> > - For car racing I meant specifically how the states are input into the GMM and not the policy. Do you use a CNN as a preprocessing step before the states are passed to the GMM/do you use the agent's CNN, or do you just flatten the obs and pass it to the GMM?
> >
> > - Could you please expand a bit more on how you can incorporate a temporal relationship between states? Naively the thing to do would be to effectively frame stack to get an augmented state space. Are there other ways?
> > - Relatedly, the policy's representation does not have to be the same as GENIE's, right? So you could framestack for one but not the other?
> >
> >
> > - Then regarding the increased state coverage. If I understand correctly, you are saying the order & diversity matters, and not diversity alone? In that case I think rephrasing the discussion in l297-l300 would be beneficial, as I at least did not get that impression from reading it.

---

> > > ### Author Response · Authors · 2024-08-09
> > > **Replying to Reviewer jT1H**
> > >
> > > Thank you for the very prompt response!
> > >
> > > **Q: For car racing I meant specifically how the states are input into the GMM and not the policy. Do you use a CNN as a preprocessing step before the states are passed to the GMM/do you use the agent's CNN, or do you just flatten the obs and pass it to the GMM?**
> > >
> > > A: For the Car Racing domain, we are also using the CNN-preprocessed states for the GMM.
> > >
> > > **Q: Could you please expand a bit more on how you can incorporate a temporal relationship between states? Naively the thing to do would be to effectively frame stack to get an augmented state space. Are there other ways?**
> > >
> > > A: There is active research regarding incorporating temporal information between states. Frame-stacking as you mentioned, is a simple method to incorporate temporal information for pixel environments. Other general methods would be to include recurrent or attention layers. For more specific hierarchical RL methods, temporal abstractions via decomposing sequences into simpler sub-tasks and operating over different temporal scales can be considered.
> > >
> > > **Q: Relatedly, the policy's representation does not have to be the same as GENIE's, right? So you could framestack for one but not the other?**
> > >
> > > A: That is correct. The representation used in GENIE is flexible and really depends on the degree of abstraction/specificity the developer is interested in regarding novelty in states.
> > >
> > > **Q: Then regarding the increased state coverage. If I understand correctly, you are saying the order \& diversity matters, and not diversity alone? In that case I think rephrasing the discussion in l297-l300 would be beneficial, as I at least did not get that impression from reading it.**
> > >
> > > A: Not quite. Our use of the word "order" in different contexts can admittedly be confusing, but we appreciate the opportunity to clarify. In Lines 184-187, we expressed a key strength of GENIE being its prioritization of diversity in individual induced experiences of the environment, independent of the order in which they are presented. In this context, "order" is **intra**-environment and refers to the **sequence of experiences (state-action pairs)** presented by a **single environment**.
> > >
> > > For our previous response regarding the differences in ACCEL and PLR, we were talking about "order" in the **inter**-environment context, more specifically how the **UED algorithm's teacher** presents the **sequence of environments** (i.e. the curriculum). GENIE enhances the diversity of individual experiences throughout all the environments in the entire curriculum but the sequence in which these diverse environments is presented in the curriculum depends on the underlying algorithm. The reason why ACCEL performs better than PLR is because the former's bootstraps its curriculum with simple levels (e.g. empty mazes) and gradually increases in complexity via minor mutations but the latter simply uses domain-randomized levels throughout.
> > >
> > > We hope the above clarifies the confusion. If not, we're happy to provide more specific examples and clarifications.

---

> > > > ### Comment · Reviewer_jT1H · 2024-08-10
> > > >
> > > > Thank you.
> > > >
> > > > > A: For the Car Racing domain, we are also using the CNN-preprocessed states for the GMM.
> > > >
> > > > For this, could you foresee a problem with the CNN's representation becoming stale? I.e., as the agent learns, the representations of the same levels would change, inducing a sort of shift. So a highly novel state could come from a previously played level, the novelty being from the updated representation.
> > > >
> > > > > A : Not quite. Our use of the word "order" in different contexts can admittedly be confusing,
> > > >
> > > > Thanks for this. Clarifying this in the updated manuscript would be helpful.

---

> > > > > ### Author Response · Authors · 2024-08-10
> > > > > **Replying to Reviewer jT1H**
> > > > >
> > > > > Once again, thank you for your engagement in this discussion phase and the following should clarify your questions:
> > > > >
> > > > > **Q: For this, could you foresee a problem with the CNN's representation becoming stale? I.e., as the agent learns, the representations of the same levels would change, inducing a sort of shift. So a highly novel state could come from a previously played level, the novelty being from the updated representation.**
> > > > >
> > > > > A: You raised a very astute point and we have considered this previously. This “representational shift” issue is circumvented by storing the raw image states (instead of learned representations) in GENIE's buffer and getting the updated representations from the CNN model before GMM fitting. In retrospect, if the designer does choose to store learned representations for convenience, GENIE's choice of using a FIFO buffer would also naturally mitigate this representational shift.
> > > > >
> > > > > **Q: Clarifying this in the updated manuscript would be helpful.**
> > > > >
> > > > > A: Definitely, thanks for highlighting this!

---

> > > > > > ### Comment · Reviewer_jT1H · 2024-08-12
> > > > > >
> > > > > > Thanks. I have updated my score to 7.

---

> > > > > > > ### Author Response · Authors · 2024-08-12
> > > > > > > **Thank you Reviewer jT1H**
> > > > > > >
> > > > > > > It has been a pleasure engaging with you during this discussion phase. We are pleased to have addressed your questions thoroughly and appreciate your increased confidence in our paper.

---

### Official Review · Reviewer_kpCs · 2024-07-11

**Soundness:** 3
**Presentation:** 3
**Contribution:** 3
**Rating:** 6
**Confidence:** 4

**Summary:**

This paper focus on the unsupervised environment design (UED) problem, whereby a student trains in an adaptive curriculum of environments proposed by a teacher. The authors propose GENIE, a method for assessing novelty of environments, which essentially means the teacher prioritizes environments with high exploration potential/info gain for the student. Experiments are conducted in three different relevant domains and the results are clear and show a reasonable gain. I thus favor acceptance.

One thing to note is the general idea of choosing environments based on novelty (rather than regret) is itself a new idea, thus, the authors do not need to focus as much on the use of GMMs, which may be limiting.

**Strengths:**

* The method makes sense as it is essentially an intrinsic reward for the policy, i.e. selecting environments where the agent will have higher information gain.
* The empirical results are strong, with the method improving both PLR and ACCEL in three different domains.
* The use of the evaluation protocol from Agarwal et al is always refreshing.
* Ablations are sensible and easy to follow.

**Weaknesses:**

* There is a strong assumption that all transitions are independent, which is not always true. What if there is an environment where an agent needs to conduct some initial behavior (e.g. finding a key) and then using it to act later (e.g. opening a door)? It just feels like the method is designed based on the toy environments we use in RL research, and is not actually scalable for larger more complex domains in the future.
* Hate to be that person, but the name GENIE is taken by multiple works already and recently by Bruce et al (2024) in a highly relevant paper. It would be recommended to find a more relevant acronym or simpler name.
* The arrows in figure 2 are too small.

**Questions:**

What other approaches could be used to assess novelty at larger scale? Can you show any examples of levels that have high novelty but low regret, and turn out to be useful training levels for the agent?

**Limitations:**

Discussed in the Appendix.

---

> ### Author Rebuttal · Authors · 2024-08-06
>
> We sincerely appreciate the reviewer's feedback and the recognition of the key strengths of our method. The following clarifications will effectively address your concerns and further enhance our paper:
>
> ### Weaknesses:
>
> **Q: There is a strong assumption that all transitions are independent, which is not always true. What if there is an environment where an agent needs to conduct some initial behavior (e.g. finding a key) and then using it to act later (e.g. opening a door)? It just feels like the method is designed based on the toy environments we use in RL research, and is not actually scalable for larger more complex domains in the future.**
>
> A: Temporal information can be accounted for by using the augmented states and it should be addressed by the underlying RL mechanism (hierarchical RL, constrained RL, etc.). We would like to highlight that GENIE's design to focus on individual *(s, a)* transitions and not entire trajectories is agnostic to these augmented state representations used by the policy. On this note, we would make this advantageous characteristic of GENIE's design clearer in the revised manuscript."
>
>
> **Q: Hate to be that person, but the name GENIE is taken by multiple works already and recently by Bruce et al. (2024) in a highly relevant paper. It would be recommended to find a more relevant acronym or simpler name.**
>
> A: Acknowledged in global rebuttal.
>
> **Q: The arrows in figure 2 are too small.**
>
> A: Thank you and we have fixed this in the revised manuscript.
>
> ### Questions:
>
> **Q: What other approaches could be used to assess novelty at larger scale?**
>
> A: With regards to assessing novelty at a large scale, Section B in the appendix touches on how dimensionality-reduction techniques can be paired with GMMs for better scalability to higher-dimensional domains. As you pointed out, there is no need to restrict ourselves to GMMs. However, on the flip side, we showed that such a simple yet general method for quantifying novelty and combining it with regret could result in significant empirical gains. GENIE's main contribution lies in demonstrating the importance of novelty in UED and highlighting how it complements minimax regret. That opens the door to future UED research to look into incorporating novelty into their curriculum.
>
> **Q: Can you show any examples of levels that have high novelty but low regret, and turn out to be useful training levels for the agent?**
>
> A: Addressed in the global rebuttal (see section "Low Regret but High Novelty Levels Provide Interesting Experiences", i.e. Figure 2's explanation)

---

> > ### Comment · Reviewer_kpCs · 2024-08-10
> > **No change to score**
> >
> > Thank you for the rebuttal. I don't see much scope to increase my score, I think the paper has sufficient merit to be accepted. If it is accepted please include a discussion on scaling the approach in the future work/conclusion. This could even be by combining it with an environment generator like Bruce et al's Genie :)

---

> > > ### Author Response · Authors · 2024-08-11
> > > **Thank you Reviewer kpCs**
> > >
> > > Thank you for considering our rebuttal and for your continued confidence in our paper.
> > >
> > > > If it is accepted please include a discussion on scaling the approach in the future work/conclusion. This could even be by combining it with an environment generator like Bruce et al's Genie :)
> > >
> > > Absolutely, we are enthusiastic about the potential for future work to further develop and broaden our novelty-driven autocurricula approach. Thank you once again for your time and support.

---

### Official Review · Reviewer_KtoT · 2024-07-13

**Soundness:** 3
**Presentation:** 3
**Contribution:** 2
**Rating:** 6
**Confidence:** 3

**Summary:**

This paper proposes using novelty quantification in an unsupervised environment design for training a more generalizable policy. Built on an intuition that environments with unfamiliar states are novel environments, their proposed algorithm uses Gaussian mixture models to allow an RL agent to explore novel environments. The authors of this paper compare their proposed method in various benchmarks against multiple baselines to show empirical improvement in performance.

**Strengths:**

This is a well-written paper with a structure that is easy to follow. The key concepts are simple and easy to understand.

**Weaknesses:**

- The idea of using unfamiliar states to quantify uncertainty seems similar in concept to the curiosity-driven approaches in RL. However, this paper does not address the relevant literature. It would be much better if the authors could explain how their idea fits in with the findings and theory in curiosity-based RL approaches and what the paper contributes to this avenue of thinking.

- When it comes to UED in curriculum learning to train a more generalizable policy, there is Genetic Curriculum (Song et al, 2022) which also uses UED in curriculum learning to train a generalizable policy. Since the paper was also evaluated on the BipedalWalker and BipedalWalkerHardcore environments, it would be better to compare and contrast how GENIE with Genetic Curriculum.

- GENIE uses a fixed window FIFO buffer, but would this be able to reach an equilibrium? For example, if the agent explores level set A at the expanse of forgetting level set B, and goes back to exploring level set B at the expanse of forgetting C, and so on, would agents trained by GENIE converge to a steady-state behavior?

- Finally, there are some minor typos and editorial mistakes. For example, line 63, PLR, and ACCEL are mentioned without explaining what those are.

**Questions:**

The questions I hope to be addressed by the authors are the ones listed above. 1) Explanations on how this paper fits in with curiosity-based approaches and the GENIE's contributions, 2) explanations and comparisons with Genetic Curriculum, and 3) Will the agents trained by GENIE reach a steady-state equilibrium?

**Limitations:**

The authors has addressed the limitations of this paper.

---

> ### Author Rebuttal · Authors · 2024-08-06
>
> We appreciate the reviewer's time and valuable feedback. We believe the concerns raised generally pertain to broader problems in RL and are not inherent problems of GENIE. The following clarifications will explain our stance and strengthen the case for our paper. We kindly request the reviewer to reconsider the score, given that the issues raised do not detract from the contributions of GENIE and the promising direction of novelty-driven autocurricula:
>
> ### Questions
>
> **Q: Using unfamiliar states to measure uncertainty is conceptually similar to curiosity-driven approaches in RL. However, the paper doesn't review the relevant literature. It would be helpful if the authors explained how their idea aligns with curiosity-based RL theories and what their paper adds to this field.**
>
> A: We appreciate the reviewer's astute observation regarding the conceptual similarity between our approach and curiosity-driven RL. While we acknowledge that our original manuscript should have addressed this relationship more explicitly, we're grateful for the opportunity to clarify these connections and distinctions. Indeed, both curiosity-driven RL and our UED approach leverage the concept of novelty or unfamiliarity to guide learning. However, they differ significantly in their application and theoretical foundations. Curiosity-driven learning literature is build on prioritising interesting experiences in a **static environment** [1], or across a set of **predefined tasks** [2]. Meanwhile, UED is focussed on **generating environments** that are interesting/useful for learning. UED shapes the learning curriculum itself rather than the exploration strategy within environments. This is analagous to the difference between Prioritized Experience Replay [3] from traditional RL and Prioritized Level Replay [4] from UED. The former is an "inner-loop" method to prioritize past experiences for training and the latter is an "outer-loop" method that uses past experiences to inform the collection/generation of future experiences. In the same vein, curiosity-driven learning is focussed on prioritizing novel experiences for policy updates but GENIE is focussed on generating/curating levels that can induce these novel experiences. This fundamental difference in purposes means that theoretical and empirical comparison between curiosity-driven approaches and GENIE is not as direct. As such, we focussed our attention mostly towards current novelty measures in UED literature, which we are of more relevance. Still, we thank the reviewer for making this observation as the general audience would also appreciate clarity on this matter. We have included a short section clarifying the distinctions between curiosity-driven learning and GENIE in the appendix of the revised paper.
>
> [1] Pathak, D., Agrawal, P., Efros, A. A., & Darrell, T. (2017). Curiosity-driven exploration by self-supervised prediction.
>
> [2] Burda, Y., Edwards, H., Pathak, D., Storkey, A., Darrell, T., & Efros, A. A. (2018). Large-Scale Study of Curiosity-Driven Learning.
>
> [3] Schaul, T., Quan, J., Antonoglou, I., & Silver, D. (2016). Prioritized experience replay.
>
> [4] Jiang, M., Grefenstette, E., & Rocktäschel, T. (2021). Prioritized level replay.
>
>
> **Q: In UED for curriculum learning to train generalizable policies, Genetic Curriculum (Song et al., 2022) also employs UED. Since both papers were evaluated on BipedalWalker and BipedalWalkerHardcore, it would be useful to compare GENIE with Genetic Curriculum.**
>
> A: Thanks for pointing to the work by Song et al. (2022). The other UED papers had not referenced this work or compared against this work. This is likely due to the fact that their problem definition in their paper differs from the *Underspecified Partially Observable Markov Decision Process* (UPOMDP) setting in UED and there is not many parallels other than sharing a commonly-used domain. Also, the original POET [5] paper which they compare against used a 5D-BipedalWalker domain (we used 8D), and it is not clear in their paper which domain they use. To the best of our knowledge and thorough search, their code repository is not publicly accessible and we are unable to replicate their work to compare the results.
>
> [5] Wang, R., Lehman, J., Clune, J., \& Stanley, K.O. (2019). Paired Open-Ended Trailblazer (POET): Endlessly Generating Increasingly Complex and Diverse Learning Environments and Their Solutions.
>
> **Q: GENIE uses a fixed window FIFO buffer. Will this achieve equilibrium? For instance, if the agent cycles through level sets A, B, and C, potentially forgetting each set as it explores others, will agents trained by GENIE converge to steady-state behavior?**
>
> A:  Your description of a potential "mode collapse" behavior is reasonable. However, this seems to be pointing at the general negative effects of catastrophic forgetting within the broader RL literature and function approximation methods. While GENIE does not explicitly guarantee convergence to a steady-state behavior with respect to novelty, it's important to note that this challenge is not unique to our approach. The leading approaches, PLR and ACCEL are also unable to provide robustness guarantees against such oscillatory exploration patterns. It would be interesting to start having conversations on how we can bridge insights from "continual learning" and "stability-plasticity dilemma" literature with UED, and see how minimax-regret and GENIE's novelty could possibly circumvent the negative effects of catastrophic forgetting.
>
> **Q: Finally, there are some minor typos and editorial mistakes. For example, line 63, PLR, and ACCEL are mentioned without explaining what those are.**
>
> A: Thank you for pointing this out, we have fixed this in the revised manuscript.

---

> > ### Comment · Reviewer_KtoT · 2024-08-10
> >
> > The reviewer thanks the authors for a well-detailed rebuttal. The reviewer has updated the scores accordingly.

---

> > > ### Author Response · Authors · 2024-08-11
> > > **Thank you Reviewer KtoT**
> > >
> > > We sincerely appreciate your engagement in the review process. We are glad that our rebuttal has strengthened your confidence in our paper.

---

### Official Review · Reviewer_qR9x · 2024-07-13

**Soundness:** 3
**Presentation:** 3
**Contribution:** 3
**Rating:** 7
**Confidence:** 4

**Summary:**

This paper proposes adding a domain-general metric for promoting novelty to state of the art UED methods in order to help the environment generator better explore and cover the space of environments.  The novelty bonus is based on a surprise of a (state, action) pair model which is a learned Gaussian mixture model.

**Strengths:**

The method is clearly useful and easy to implement, fixing a limitation of existing UED approaches which are known to miss modes in the space of levels.  Empirically the method seems to perform strongly, matching or exceeding existing methods. This is a valuable direction to pursue as it is important for the UED community to have a sense for how novelty effects performance, and the approach addresses a well-known flaw in existing UED approaches.

**Weaknesses:**

The method would be more convincing if the generated levels for each method where visualised and compared directly.  Ideally one could demonstrate that there are a motif of level being generated which were not being generated before.  For instance, displaying the distribution over stump hight late in training of ACCEL and ACCEL-GENIE to show that ACCEL-GENIE has a more even coverage of the space, and sampling a few levels randomly from each so that this can be visually confirmed.

Similarly, the paper would be much improved if the mechanism behind the results was dug into in more depth. If the theory is that the diversity term results in kinds of high-regret levels being presented which were left out of the buffer previously, it would be ideal to demonstrate that this happens directly.

On line 258 results are over-claimed, since ACCEL-GENIE is within the margin for error of ACCEL in all, or nearly all, of the environments. This claim should be corrected. I don't think this is essential for acceptance of the paper, as the minigrid benchmark appears pretty close to being saturated, and it serves largely as an MNIST for the field. The empirical results in the other two environments stand on their own.

#### Clarity:
It would be good to have error bars in Table 1
The plots in Figure 7 should be fixed so the colors of the same method match
There is a stray parenthesis on line 303
Equation 1 shows the PAIRED objective "optimized" for deterministic domains, the PAIRED objective as a direct comparison between the two expected returns is canonical.

I would also suggest reconsidering the name, as another prominent environment-generating model named GENIE has been released recently. I expect that, in talking about this work, people will want to talk about how the two could be combined, which could get quite confusing.

In the abstract I think it is not clear that novelty, on it's own, is critical for agent's generalisation ability. Two situations which are different but not in a way that is relevant for the task and are processed by the network in the same way are effectively the same, and would not effect generalisation ability.

It also appears that in a few places the "underspecified" in UPOMDP is taken to mean that there is a one-to-many mapping between parameters and environments. In general this is not the case, as the minigrid environment has a one-to-one mapping between parameters and environments, the "underspecified" simply means that these parameters are not given by the designer. It is fair to want UED algorithms to work even when there is such a one-to-many mapping, but it is not required as part of the problem formulation.

**Questions:**

What sort of qualitative difference do you see between ACCEL-GENIE and ACCEL levels?

**Limitations:**

See weaknesses.

---

> ### Author Rebuttal · Authors · 2024-08-06
>
> We sincerely appreciate the reviewer's insightful comments and recognition of the valuable direction of novelty-driven autocurriculum that GENIE is pushing for the UED field. Also, it is always refreshing to engage with a reviewer who has in-depth knowledge of UED. The new results within the global rebuttal and our following clarifications would be of interest to you and strengthen the case for our paper:
>
> ### Weaknesses
> **Q: The method would be more convincing if the generated levels were visualized and directly compared. Ideally, demonstrate that new motifs are being generated. For example, display the distribution over stump height late in training of ACCEL and ACCEL-GENIE to show more even coverage by ACCEL-GENIE, and randomly sample a few levels from each for visual confirmation.**
>
> A: Answered in global rebuttal, refer to the section "GENIE Introduces Level Complexity" explanation.
>
> **Q: The paper would be improved by exploring the mechanism behind the results in more depth. If the theory is that the diversity term introduces high-regret levels previously excluded from the buffer, it should be demonstrated directly.**
>
> A: This is something we have thought about while working on GENIE. The main struggle with trying to demonstrate this is that both the regret and novelty metric are inherently policy-dependent. As such, it is impossible to directly measure whether prioritising or not prioritising a level via GENIE would lead to discovering a higher regret level down the road because the divergence in realities would result in different policies with non-commensurable subjective regrets. However, one way we can indirectly measure this is by observing whether a greedy selection of high regret levels (as in ACCEL and PLR) or a balanced regret-novelty prioritization (as in ACCEL-GENIE and PLR-GENIE) results in better cumulative/mean regret in the replay buffer across the training horizon. The section "Prioritizing Novelty Actually Increases Regret" in our global rebuttal demonstrates that GENIE actually results in higher regret across the replay buffer in the Car-Racing domain despite not directly optimising for it.
>
>
> **Q: On line 258, results are overstated, as ACCEL-GENIE's performance is within ACCEL's margin of error in nearly all environments. This claim should be corrected. This isn't essential for paper acceptance, as the minigrid benchmark is nearly saturated and serves as an MNIST for the field. The empirical results in the other two environments are sufficient.**
>
> A: We have amended line 258 to exclude the remark on ACCEL-GENIE's outperformance over its predecessor but the point on PLR-GENIE's clear improvement over PLR still holds.
>
> **Q: Add error bars to Table 1. Ensure matching colors for the same method in Figure 7. Fix the stray parenthesis on line 303. Equation 1 should reflect the canonical PAIRED objective, comparing the two expected returns directly, rather than being "optimized" for deterministic domains.**
>
> A: Thank you for the attentiveness to detail, we have addressed this in our revised manuscript.
>
> **Q: Consider renaming the model since another prominent environment-generating model named GENIE has been released recently. This will avoid confusion when discussing how the two could be combined.**
>
> A: Addressed in global rebuttal.
>
> **Q: In the abstract I think it is not clear that novelty, on its own, is critical for agent's generalisation ability. Two situations that are different but not in a way that is relevant for the task and are processed by the network in the same way are effectively the same, and would not affect generalisation ability.**
>
> A: We are actively refining our abstract to more effectively advocate for the benefits of novelty-driven autocurricula methods, while being mindful of the brevity throughout this rebuttal process. Regarding the notion that "different situations processed similarly by the network do not affect generalization ability," we believe this holds true primarily for fixed curriculum methods but not necessarily for autocurricula methods. We hope to have a constructive discussion about this.
>
> Although the two situations present differently but provide similar learning experiences (with relation to the task) at the current moment, they both independently inform the collection/generation of future environments. The fact that these environments behave differently (in state-action distribution) indicates the potential that they can lead to totally different and novel environments down the training horizon, especially with mutation-based methods. Therefore, even if these scenarios yield similar learning signals initially, prioritizing them for the unique state-action distributions they cover via GENIE remains beneficial for generalisation. This is an exciting discussion which can be correspondingly highlighted in our revised manuscript.
>
> **Q: It also appears that in a few places the "underspecified" in UPOMDP is taken to mean that there is a one-to-many mapping between parameters and environments. In general this is not the case, as the minigrid environment has a one-to-one mapping between parameters and environments, the "underspecified" simply means that these parameters are not given by the designer. It is fair to want UED algorithms to work even when there is such a one-to-many mapping, but it is not required as part of the problem formulation.**
>
> A: We acknowledge that the one-to-many mapping between free parameters and environments, while common, is not a mandated characteristic of UED. We have amended the manuscript to reflect that nuance more clearly (e.g., "entails there is a one-to-many mapping" is revised to "possibly entails a one-to-many mapping").
>
>
> ### Questions
> **Q: What sort of qualitative difference do you see between ACCEL-GENIE and ACCEL levels?**
>
> A: Answered in global rebuttal, refer to the section "GENIE Introduces Level Complexity".

---

> > ### Comment · Reviewer_qR9x · 2024-08-10
> > **Response to Rebuttal by Authors**
> >
> > Thank you for your response. The added visualisations increase my confidence in the method, so I will be increasing my score accordingly.
> >
> > > As such, it is impossible to directly measure whether prioritising or not prioritising a level via GENIE would lead to discovering a higher regret level down the road because the divergence in realities would result in different policies with non-commensurable subjective regrets.
> >
> > One test that could show this effect would be if the policies from both realities found the levels from the GENIE reality to have higher regret. I agree if this was not the case it would not be negative evidence.
> >
> > Another good measure would be to fix a policy (and freeze it so it does not train) and run both to see which finds higher regret levels.

---

> > > ### Author Response · Authors · 2024-08-11
> > > **Thank you Reviewer qR9x**
> > >
> > > Thank you for your engagement and providing such technical feedback. We truly appreciate your confidence in our work.
> > >
> > > > One test that could show this effect would be if the policies from both realities found the levels from the GENIE reality to have higher regret. I agree if this was not the case it would not be negative evidence.
> > >
> > > > Another good measure would be to fix a policy (and freeze it so it does not train) and run both to see which finds higher regret levels.
> > >
> > > Also, thanks for suggesting these creative experimental setups. We'll begin implementing these experiments and will incorporate the results in our revised manuscript.

---

### Author Rebuttal · Authors · 2024-08-06

# Global Rebuttal

## 1. Addressing GENIE's Name
There is a general consensus among the reviewers that "GENIE" might be confused with the method recently introduced by Bruce et. al. (2024). We agree that it would probably be wise to choose a different name for the framework, allowing both important works to receive their own deserved spotlights. However, in the meantime, we will still use the acronym "GENIE" when referencing our framework during this rebuttal phase to make things less confusing for everyone. We will definitely come up with a different name and implement it across the revised version of the paper.

## 2. Special Thanks to All Reviewers
We would like to express our gratitude to all the reviewers for their time in providing constructive evaluations and generally positive reception of our work.

## 3. New Results and Explanation
We present new results that are collected in response to the reviewers' comments (please refer to the attached 1-page PDF). Note that due to limitations in time and computation, we were only able to selectively run experiments.

### 3.1 GENIE Introduces Level Complexity
Figure 1 presents the difficulty composition (introduced by POET (Wang et al., 2019) of replayed levels for ACCEL and ACCEL-GENIE over various training intervals (based on metrics defined in Table 1).  It is clear that ACCEL predominantly favors "Easy" to "Moderate" difficulty levels. In contrast, ACCEL-GENIE increasingly incorporates "Challenging" levels into its replay set over time. This difference highlights the benefits of integrating GENIE's novelty metric into the level replay selection criteria.

The disparity in level difficulty distribution between ACCEL and ACCEL-GENIE is a critical factor in their observed performance differences. ACCEL's training curriculum tends to remain within a comfort zone, where levels with high regret (approximated by TD-error) are greedily selected. This approach constrains the student to a limited subset of simplest environments where it can minimize its maximum prediction error. However, this narrow focus limits the student's ability to generalize, as it minimizes exposure to more complex scenarios. On the other hand, ACCEL-GENIE’s incorporation of the novelty metric actively selects more challenging levels. This strategy pushes the student beyond its comfort zone, exposing it to unfamiliar and more challenging environment parameters (e.g. higher stump heights and wider pit gaps). As a result, the student is forced to explore a broader state-action space, enhancing its robustness to out-of-distribution scenarios and leading to the discovery of higher regret levels.

Note that our figure differs from Figure 11 in Parker-Holder et al. (2022) which shows the difficulty distribution of the levels **generated and added into the buffer**, but not the actual levels selected by the teacher for the student to **replay/train on**. On that note, this also demonstrates that GENIE remedies an inefficiency in the original ACCEL algorithm, which is the fact that the mutation-based generation constantly produces high complexity levels ("Challenging" and above) but none are actually selected to train the student.

At the moment, we currently lack statistics on the difficulty composition of levels replayed by PLR and PLR-GENIE but their similar performances suggest that the difficulty compositions are likely comparable.

[1] Parker-Holder, J., Jiang, M., Dennis, M., Samvelyan, M., Foerster, J.N., Grefenstette, E., \& Rocktaschel, T. (2022). Evolving Curricula with Regret-Based Environment Design.

### 3.2 Low Regret but High Novelty Levels Provide Interesting Experiences
Next, we visually present the effect of the novelty metric on Minigrid levels in the level replay buffer of PLR-GENIE by ablating regret. Specifically, we highlight levels that feature the lowest regret (bottom 10) yet exhibit the highest novelty (top 10); these are showcased in the first row of Figure 2. Conversely, levels that score within the lowest 10 for both regret and novelty are displayed in the second row of the same figure.

Visually, we can observe that levels with high novelty and low regret present complex and diverse scenarios that challenge the student. In contrast, the levels displayed in the second row, characterized by low regret and low novelty, often resemble simple, empty mazes that offer limited learning opportunities.

While it is not feasible to present every example level here, the contrast between the two groups is stark. Levels selected based on low regret but high novelty are significantly more varied and intricate than those chosen for their low novelty, despite both groups having low regret scores. This demonstrates that incorporating novelty alongside regret in the selection process enhances the ability to identify levels that present more interesting trajectories (experiences) to the student for learning.

### 3.3 Prioritizing Novelty Actually Increases Regret
Finally, Figure 3 shows the mean, median and summed regret in the level replay buffer of PLR$^\perp$ and PLR-GENIE across the training horizon. Surprisingly, PLR-GENIE results in comparable/slightly greater levels of regret across the training distribution despite not directly optimising for it. This observation demonstrates that prioritising novelty in the levels can actually lead to higher regret levels being discovered.

---

### Decision · Program_Chairs · 2024-09-25

**Decision:**

Accept (oral)

**Comment:**

This paper introduces an algorithm for exploring environment spaces efficiently based on a new novelty metric.
The authors provide compelling experimental results across a range of environments and were also able to address the concerns of the reviewers through the rebuttal phase.
This resulted in a number of score increases and an strong consensus that the paper should be accepted.
Given the relevance of the work I believe it would be great for this work to be highlighted as an oral at the conference.